# 100 INSTANCES IS ALL YOU NEED: PREDICTING LLM SUCCESS BY TESTING ON A FEW INSTANCES

## ABSTRACT

Predicting if LLMs will succeed on individual task instances (i.e., prompts) is essential to ensure their reliability in high-stakes applications. To do so, we can evaluate a LLM on a set of instances and train an *assessor* to predict its performance. However, this requires evaluating each new LLM on sufficiently many instances. In this work, we build a *generic assessor* predicting the performance of any LLM on an instance by using the LLM's performance on a small set of reference instances and the features of the considered instance. In practice, we make use of existing evaluation results to extract the representative instances and train the assessor. Thus, the performance of a new LLM can be predicted by only testing it on the reference instances, leveraging the information contained in other LLMs' evaluations. We conduct empirical studies on HELM-Lite and KindsOfReasoning, a new collection of existing reasoning datasets that we introduce, where we evaluate all instruction-fine-tuned OpenAI models until `gpt4-0125-preview`. We find that a few instances (around 100) are enough to achieve predictive power comparable to the LLM-specific assessors trained on the complete set of several thousand instances. Interestingly, randomly selecting the reference instances performs comparably to the advanced selection methods we tested. Finally, we identify a sharp drop in predictive power of the generic and specific assessors in out-of-distribution scenarios, suggesting that the inherent predictability of LLMs is low.

## 1 INTRODUCTION

Large Language Models (LLMs) are being used as components of multiple services and products, such as agents performing general computer tasks (Kim et al., 2024), performing ML experiments (Huang et al., 2024), and even operating unmanned aerial vehicles (Javaid et al., 2024). These systems typically query an LLM on a specific instance (i.e., a specific prompt) of a task and use their output to determine an action. For some of these uses, it is essential to determine if the output produced by the LLM on a specific task instance is likely to be correct (or, more generally, "valid" (Zhou et al., 2023)) before the subsequent steps are executed[1]. A nascent line of research (Zhou et al., 2022; Hernández-Orallo et al., 2022; Drapal et al., 2024) is addressing this problem by developing "assessors", namely, independent modules that predict the correctness (or a continuous score) of an AI system on an instance based on features intrinsic to the latter (such as linguistic features or sentence vector embeddings). Assessors can be specific to an AI system, or "generic", in which case they also take as input features of the AI system at hand and are trained to predict the performance of different LLMs on different instances.

Meanwhile, the rate at which new LLMs are released has drastically increased. Some providers, such as OpenAI, are iteratively retiring old versions when new ones are released, forcing developers to update the LLM version used in their applications (see OpenAI (2024a)). An even larger explosion is occurring in the open-source world, fuelled by inexpensive fine-tuning techniques (Hu et al., 2022). To build an assessor specific to a new LLM version, users must evaluate it on a sufficiently large set of task instances, causing the costs to rise quickly when considering many LLM versions. On the other hand, the system information one might use to build a generic assessor, such as the number

---

[1]Notice that this cannot rely on the "ground truth" of the task instance, as that is not available in practical use cases (otherwise, there would be no need to query the LLM).

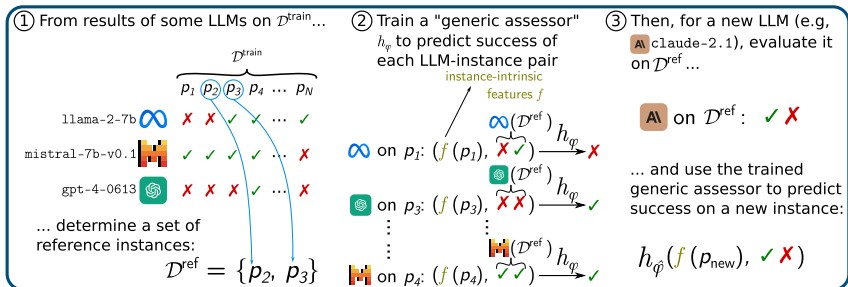

Figure 1: Our proposed pipeline for predicting the performance of a new LLM on a new instance by testing on a few instances: starting from instance-level evaluation results of a set of LLMs, a reference set of instances is extracted (1). Then, we train a "generic assessor" that predicts the performance of each LLM-instance pair, based on features intrinsic to the instance (e.g., vector embeddings) and the performance of the considered LLM on the reference set (2). The performance of the new LLM on a new instance can be predicted by evaluating the new LLM on the reference set and applying the trained generic assessor (3).

of parameters or statistics of the training data or architecture, is not standardised across LLMs and unavailable for proprietary models.

As such, this paper investigates the following question: **can we combine information across LLMs to predict the performance of a new LLM on a new instance by relying only on observational (or behavioural) features of the LLMs?** In practice, we propose to characterise each LLM by its performance on a small set of *reference instances* and to build a generic assessor using those as system features. More precisely, we first select a small set of reference instances from the labelled dataset on which past LLMs were evaluated. Then, we train the generic assessor on the concatenation of instance-specific features and the LLM-specific success vector on the reference instances. Finally, to estimate the probability of success of a new LLM on a novel instance, it suffices to evaluate the former on the reference instances, concatenate its performance to the features of the instance, and apply the trained generic assessor. See Fig. 1 for a graphical representation of this procedure.

In our empirical studies, we rely on HELM-Lite (Liang et al.), which provides instance-level results for 30 LLMs from different providers (at the time we conducted our experiments), and a collection of previously existing datasets we introduce, named "KindsOfReasoning", on which we evaluated the full set of instruction-following models from OpenAI until `gpt4-0125-preview`. We only consider tasks with binary correctness score (therefore discarding the datasets in HELM-Lite that do not satisfy this) and thus build binary assessors.

We train specific assessors using different prompt features and find that OpenAI embeddings (OpenAI, 2024b) lead to better or comparable in-distribution performance than simpler methods such as `Word2vec` (Mikolov et al., 2013), `FastText` (Bojanowski et al., 2017), and n-grams. Subsequently, we build generic assessors using various methods to select the reference instances and combine the performance on these with the instance-specific features. When predicting performance on instances with the same distribution as those used to train the generic assessor, we find the latter to perform comparably to the specific assessors, which require the LLM to be evaluated on many more instances. Additionally, we find that a random selection of reference instances performs as well as the advanced selection methods we tested. However, in out-of-distribution scenarios, the predictive power of all assessors declines significantly, indicating a lack of general predictability in LLMs.

In essence, the main contributions of our work are the following:

- We propose a framework combining evaluation results across LLMs to predict the performance of a new LLM by only evaluating it on a a small set of reference instances.
- We study the performance of various methods for selecting the reference instances and combining their performance with instance-specific features to build the generic assessor.
- Finally, we introduce *KindsOfReasoning:* A new compilation of existing datasets testing various kinds of reasoning and release the raw outputs of a large number of models from OpenAI.

## 2 RELATED WORK

### 2.1 INSTANCE-LEVEL PREDICTION OF SUCCESS OF AI SYSTEMS

Zhou et al. (2023) advocates for the importance of instance-level success predictions for AI systems and coins the term "predictable AI"; in particular, they highlight how ensuring predictability should be prioritised over increases in average performance for high-stakes use cases. Following this motivation, Hernández-Orallo et al. (2022) introduces the concept of an *assessor model*, which accompanies an ML system and estimates the probability of success of the system on individual instances. In particular, an assessor can be trained on the evaluation results of the ML system on test data (i.e., which has not been used for training the ML system). In similar spirit, Drapal et al. (2024) combines novelty detection and meta-learning to reject instances where a ML system is likely to fail. Other similar approaches are described in Section 4 of Hendrickx et al. (2024). Zhou et al. (2022) shows how a smaller LLM can be used to predict the performance of a bigger LLM on individual instances without passing the latter through the model. They also find it possible to reject almost half of the failure cases before running much larger LLMs, resulting in a significant saving of compute. Finally, an application of success prediction is "routing" between different LLMs, as explored in Shnitzer et al. (2023) and Hu et al. (2024).

### 2.2 PREDICTABILITY OF AGGREGATED BENCHMARK SCORES FROM LLM FEATURES

Two works (Ye et al., 2023; Owen, 2024) studied the extent to which an LLM's aggregate performance on BIG-Bench tasks (Srivastava et al., 2022) can be predicted using information on the LLM such as number of parameters or the amount of used compute. In contrast, our work does not rely on these quantities, which are often unavailable, instead characterising LLMs according to their performance on reference samples. Moreover, while these works focus on predicting aggregate performance, our work and the ones mentioned in the previous subsection provide instance-level predictions for new unlabelled instances.

### 2.3 EXTRACTING LLM-SPECIFIC FEATURES FROM EXISTING EVALUATIONS

Recently, Ruan et al. (2024) built "observational scaling laws" that link performance on complex downstream tasks to hypothesised latent capabilities, whose values can be inferred by decomposing the performance of various LLMs on different benchmarks into components linked by a log-linear relation with compute measures for LLM training. Once this relation is established, the performance of a new model on downstream tasks can be predicted by knowing its performance on simple benchmarks and its compute cost. Their work is similar to ours in determining LLM-specific features by using evaluation results of multiple LLMs and using them to predict the performance of a new LLM. However, we aim to predict the performance of the new LLM on a novel individual instance by evaluating on as few instances as possible, while Ruan et al. (2024) instead aims to avoid the cost of evaluating complex downstream tasks and predict the performance on the latter from results on simple benchmarks and compute measures, which they assume to be available. Moreover, our method can be applied to predict the performance on instances for which no ground truth is available, while the simple benchmarks and the downstream tasks employed in Ruan et al. (2024) must have a grading mechanism.

### 2.4 PREDICTING PERFORMANCE BY BENCHMARK SUBSAMPLING

Several works share our motivation of reducing the number of evaluations (and hence the cost) needed to evaluate a LLM. For instance, a "Lite" version with a reduced number of tasks was introduced alongside the BIG-Bench benchmark (Srivastava et al., 2022); similarly, HELM-Lite (Liang et al.) is a revised and reduced version of HELM (Liang et al., 2022). However, both of these perform the reduction at the level of *tasks* (i.e., datasets) of which the benchmark is constituted. Instead, Vivek et al. (2024) subsample a dataset by clustering models' confidence to predict the overall accuracy on the whole dataset, while MixEval (Ni et al., 2024) extracts a subset of instances from various benchmarks which is most predictive of the performance on Chatbot Arena[2], an online platform performing pairwise comparison of LLM outputs.

---

[2] https://chat.lmsys.org/

Closer to our work is TinyBenchmarks (Polo et al., 2024), which selects informative instances from HELM-Lite and estimates the performance of a new LLM on the whole benchmark by evaluating it only on those instances. In particular, TinyBenchmarks uses Item Response Theory (IRT) on the successes of each LLM to extract a vector of item demands and LLM capabilities. Then, it uses either the item demands or the raw LLM success on each instance to build a representative subset of instances by clustering the items and taking the cluster centroids. Similarly to our approach, a new LLM is then only evaluated on the representative subset; however, in contrast to our work, they aim to predict the aggregate score on the benchmark, while we predict instance-level performance. In practice, their IRT method provides instance-level predictions, but these predictions are limited to instances on which previous LLMs have been evaluated (as this is necessary to obtain the item demands), which requires access to the ground truth. In contrast, our approach relies on "intrinsic" (model-agnostic) features of the instances (alongside the performance on the reference samples, see Fig. 11), thus making it applicable to new instances with unknown ground truth, as the trained assessor does not require any information beyond the intrinsic features of test instances.

A similar work to Polo et al. (2024) is `metabench` (Kipnis et al., 2024), which considered 6 different datasets, and performed a two-step procedure (random sampling for each dataset, followed by item selection based on the Fisher information matrices of IRT item parameters) to extract a small set of instances, the performance on which accurately predicts aggregate performance on the 6 datasets. As they fit the IRT model only the pre-selected instances, their method is unable to predict instance-level performance. Finally, despite not tackling predictability directly, Siska et al. (2024) finds that the vector of successes of different LLMs is correlated across instances belonging to 4 benchmarks, and, for one of those benchmarks, the similarity between the embeddings or a pair of instances predicts the similarity between the success vectors; this suggests that patterns in success across LLMs can be found and related to the embeddings.

## 2.5 EVALUATIONS OF REASONING IN LLMS

Burnell et al. (2023a) found reasoning to be one of three factors in the capabilities of LLMs. Indeed, reasoning in LLMs has been extensively studied: see Mondorf & Plank (2024) for a survey on LLM reasoning evaluations and Huang & Chang (2023) for a broader survey also encompassing ways to improve and elicit reasoning in LLMs.

Recently, several collections of reasoning datasets have been introduced. GLoRE (Teng et al., 2023) collects 12 logical reasoning datasets with three different types of tasks (multiple choice, natural language inference, and binary answers). Similarly, LogiGLUE (Luo et al., 2023) collects 24 datasets related to inductive, deductive and abductive reasoning, with four different types of tasks (the same ones as GLoRe and free-form question answering); they only selected datasets that do not require external domain knowledge, but some of these datasets are poorly formatted. Finally, CALM-Bench (Dalal et al., 2023) is a collection of 6 diverse tasks requiring both causal reasoning and knowledge. KindsOfReasoning, the collection we introduce combining previously existing datasets testing various kinds of reasoning, partly overlaps with each of the aforementioned collections; however, KindsOfReasoning aims to include a broader range reasoning types (logical, common sense, inductive, deductive, abductive, counterfactual, causal, analogical, spatial and arithmetic reasoning) over 22 different datasets; see Appendix B for more information on the dataset construction.

## 3 METHODOLOGY

Let us denote by $\mathcal{L} = \{m_j, j = 1, \ldots, n\}$, a set of trained LLMs. Moreover, let $\mathcal{D} = \{(p_i, y_i), i = 1, \ldots, N\}$ be a test dataset used to evaluate the performance of the LLMs, with $i$ denoting instance index, $p_i$ the input to the LLM (i.e., the prompt) and $y_i$ the target value (i.e., the expected completion by the LLM). Further, we will denote by $m_j(p_i)$ the output $m_j$ produces when given $p_i$ as input[3] and by $z_{j,i}$ a binary value indicating the "correctness" of $m_j(p_i)$ with respect to $y_i$. The correctness $z_{j,i}$ can be defined in multiple manners (for instance, exact match or whether $y_i$ is a substring of

---

[3]As LLMs are stochastic, $m_j(p_i)$ is in general a random variable, and so is $z_{j,i}$. In our empirical study, we sample the LLMs at 0 temperature, but, even so, there is still a residual amount of stochasticity, even though the reason for this is unclear (OpenAI, 2023).

$m_j(p_i)$); the most suitable manner depends on the considered task, but in general the aim of $z_{j,i}$ is to capture what a human judge would perceive as a correct answer[4].

Below, we first frame the problem of predicting the correctness $z_{j,i}$ and then discuss our framework to predict the performance of a new LLM by evaluating it on a small subset of instances.

## 3.1 Predicting success of a LLM using features intrinsic to the prompt

Let us consider a single LLM, say $m_1$; we aim to train a classifier (termed "assessor") to predict the performance $z_{1,i}$ from the prompt $p_i$. To do so, we split the test dataset $\mathcal{D}$ into different splits used to train, validate and evaluate the assessor (Hernández-Orallo et al., 2022), denoted as $\mathcal{D}^{\text{train}}, \mathcal{D}^{\text{val}}$ and $\mathcal{D}^{\text{test}}$, such that $\mathcal{D} = D^{\text{train}} \cup \mathcal{D}^{\text{val}} \cup \mathcal{D}^{\text{test}}$ and $\mathcal{D}^{\text{train}} \cap \mathcal{D}^{\text{val}} = \mathcal{D}^{\text{val}} \cap \mathcal{D}^{\text{test}} = \mathcal{D}^{\text{train}} \cap \mathcal{D}^{\text{test}} = \varnothing$. In a real-world scenario, $\mathcal{D}^{\text{test}}$ will represent instances for which we did not evaluate the considered LLM (and for which we may not have access to the ground truth); in contrast, available evaluation results are split into $\mathcal{D}^{\text{train}}$ and $\mathcal{D}^{\text{val}}$.

In practice, we can extract some numerical features $f(p_i)$ from the textual prompt $p_i$; we use "intrinsic" features, i.e. features that are defined independently of the problem at hand (such as the number of negations or the vector embeddings of the sentence). Formally, we consider a loss function $\ell$ and a family of classifiers $h_\varphi$, where $\varphi$ denotes the parameters of the classifier (for instance, the weights in a logistic regression classifier), and aim to minimise

$$\sum_{p_i \in \mathcal{D}^{\text{train}}} \ell(h_\varphi(f(p_i)), z_{1,i}) \tag{1}$$

over $\varphi$ using some optimisation algorithm; we can then select the best hyper-parameters using the performance on the validation data $\mathcal{D}^{\text{val}}$, thus selecting $h_{\hat{\varphi}}$. Now, we can predict the performance of $m_1$ on $p^{\text{new}} \in \mathcal{D}^{\text{test}}$ as $h_{\hat{\varphi}}(f(p^{\text{new}}))$ without inputing the prompt $p^{\text{new}}$ into the LLM $m_1$[5].

## 3.2 Predicting success by evaluation on reference instances

Now, consider the case in which we have previously evaluated some LLMs on $\mathcal{D}^{\text{train}}$ and $\mathcal{D}^{\text{val}}$. We are interested in predicting the performance of a new LLM, say $m^{\text{new}}$ on new instances $\mathcal{D}^{\text{test}}$. We want to leverage the information contained in the available evaluation results for previous LLMs to predict the performance of $m^{\text{new}}$ on $\mathcal{D}^{\text{test}}$ without evaluating it on the full $\mathcal{D}^{\text{train}}$ (and assuming that we do not have access to the labels in $\mathcal{D}^{\text{test}}$, which prevents us from evaluating the other LLMs on it). Thus, we build a *generic assessor*, namely a classifier that predicts the success $z_{j,i}$ from the pair $(m_j, p_i)$. In practice, we split the LLMs for which full evaluation results are available into a training and validation split $\mathcal{L}^{\text{train}}$ and $\mathcal{L}^{\text{val}}$. For each pair $(m_j, p_i) \in \mathcal{L}^{\text{train}} \times \mathcal{D}^{\text{train}}$, we concatenate the prompt-intrinsic features $f(p_i)$ with LLM-specific features $g(m_j)$ and aim to fit a classifier $h_\varphi$ that minimises

$$\sum_{m_j \in \mathcal{L}^{\text{train}}} \sum_{p_i \in \mathcal{D}^{\text{train}}} \ell(h_\varphi(g(m_j), f(p_i)), z_{j,i}) \tag{2}$$

over $\varphi$. Similarly to what we did before (Section 3.1), we use the performance of $\mathcal{L}^{\text{val}}$ on $\mathcal{D}^{\text{val}}$ to perform model selection, leading to a trained classifier $h_{\hat{\varphi}}$. Then, the performance of $m^{\text{new}}$ on an instance $p^{\text{new}} \in \mathcal{D}^{\text{test}}$ can be obtained as $h_{\hat{\varphi}}(g(m^{\text{new}}), f(p^{\text{new}}))$.

The LLM-specific features $g(m_j)$ could include statistics on the training data of $m_j$ and architectural information (for example, number of attention layers and parameters). However, the high variety of hyperparameters involved in the definition and training of LLMs and the unavailability of detailed information on proprietary models makes defining broadly informative features hard, if not impossible. To circumvent this problem, we propose to use the performance of $m_j$ on a small set of reference instances $\mathcal{D}^{\text{ref}} \subset \mathcal{D}^{\text{train}}$ as $g(m_j)$: $g(m_j) = (z_{j,i})_{i \in \mathcal{D}^{\text{ref}}}$; in this way, it is sufficient to evaluate the new LLM $m^{\text{new}}$ on $\mathcal{D}^{\text{ref}}$ to predict their performance on news instances $\mathcal{D}^{\text{test}}$. See Figure 1 for a graphical description of our method. Next, we discuss various methods to determine $\mathcal{D}^{\text{ref}}$.

---

[4]Particularly in the case of free-form question answering, it can be tricky to find a formulation that always matches what a human judge would perceive as a correct answer.

[5]This assessor is anticipative (Hernández-Orallo et al., 2022), as it does not use the output $m_1(p^{\text{new}})$ when predicting the performance; this can avoid the cost of querying the LLM if its performance on a specific input is predicted to be poor.

### 3.2.1 SELECTING THE REFERENCE INSTANCES

In order to select the most informative instances $(p_i, y_i) \in \mathcal{D}^{\text{train}}$ to form $\mathcal{D}^{\text{ref}}$, we can use information intrinsic to the instances as well as the evaluation results of $\mathcal{L}^{\text{train}}$ on $\mathcal{D}^{\text{train}}$ (while keeping aside $\mathcal{D}^{\text{val}}$ and $\mathcal{L}^{\text{val}}$ to choose the best selection method; see Section 3.2.2). In general, let us denote by $x_i \in \mathbb{R}^d$ a feature vector associated to $p_i$ and $\mathbf{X} \in \mathbb{R}^{d \times |\mathcal{D}^{\text{train}}|}$ the matrix whose columns are $x_i$. Finally, let us define $\mathbf{Z}^{\text{train}} = (z_{j,i})_{j: \, m_j \in \mathcal{L}^{\text{train}}, i: \, p_i \in \mathcal{D}^{\text{train}}}$. We attempt using the following features:

- prompt features $x_i = f(p_i)$ (not necessarily the same used to build the assessor in Sections 3.1 and 3.2).
- The binary success vector on $\mathcal{L}^{\text{train}}$, which yields $\mathbf{X} = \mathbf{Z}^{\text{train}}$ and for which $d = n_{\text{train}}$.
- The item demands obtained by applying the IRT approach in Polo et al. (2024) (discussed in Section 2), which obtains a set of item demands and LLM capabilities starting from the success matrix $\mathbf{Z}^{\text{train}}$. Thus, we set $x_i$ to be the obtained item demands, whose size $d$ can be chosen by the user (we fix this to $d = 10$ following Polo et al. (2024)).

For all possible choices of $\mathbf{X}$ described above, we use two methods to determine the reference instances: first, we apply KMeans clustering on the columns of $\mathbf{X}$. For each identified cluster, we select the instance $i$ that is closest to the cluster centroid and add it to $\mathcal{D}^{\text{ref}}$. The pre-specified number of clusters dictates the number of selected instances.

The second method is Factor Analysis (FA), which decomposes $\mathbf{X}$ into $\mathbf{X} = \mathbf{WH} + \mathbf{E}$. Here, $\mathbf{W} \in \mathbb{R}^{d \times l}$ is the loading matrix, $\mathbf{H} \in \mathbb{R}^{l \times |\mathcal{D}^{\text{train}}|}$ contains the latent factors for each sample, $\mathbf{E}$ represents Gaussian noise, and $l$ denotes the number of hidden factors. In practice, we first fit FA with a high number of factors. Then, we set $l$ to the number of eigenvalues of the correlation matrix $\mathbf{XX}^T$ that exceed 1, and we re-fit FA using the varimax rotation method (Kaiser, 1958). The reference instances are then selected by picking, for each factor $k = 1, \ldots, l$, an approximately equal number of instances with the highest values of $|H_{k,i}|$[6].

Hence, we can select $\mathcal{D}^{\text{ref}}$ using one of the three sets of features with any of the two selection methods, leading to a total of 6 possible methods, two of which (clustering on success/failures and IRT item parameters) correspond to the selection method used in Polo et al. (2024). We compare these methods with a random reference subset; moreover, we also draw 20 random reference subsets, fit an assessor using the performance on the reference instances, and pick the random subset that leads to the highest performance ("random best of 20").

### 3.2.2 CHOOSING THE BEST SETUP ON VALIDATION DATA AND PREDICTING THE PERFORMANCE OF A NEW LLM

As mentioned above, we have multiple ways to define the reference set as well as multiple choices for the intrinsic features $f$. We can also choose multiple families of classifiers $h_\varphi$ and hyperparameters of the optimisation algorithm to minimise equation 2. As such, we pick the combination of options which best predicts the performance of the validation LLMs $\mathcal{L}^{\text{val}}$ on the validation data $\mathcal{D}^{\text{val}}$. Hence, once we want to predict the performance of a new LLM $m^{\text{new}}$ on a new instance $p^{\text{new}} \in \mathcal{D}^{\text{test}}$, we only need to evaluate $m^{\text{new}}$ on $\mathcal{D}^{\text{ref}}$ and apply the trained generic assessor. In our empirical studies below, we will test each method on multiple new LLMs, which we group into $\mathcal{L}^{\text{test}}$.

## 4 EMPIRICAL STUDIES

### 4.1 DATASETS

We consider two collections of datasets in our experiments. The first is HELM-Lite (Liang et al.), a revised and reduced version of the popular HELM (Liang et al., 2022), which includes 10 different "scenarios" (i.e., datasets), some of which are stratified into sub-scenarios. Of those, we keep the scenarios and subscenarios for which the performance metric is binary, and further discard those for which different LLMs were tested with a different number of few-shot examples; the resulting

---

[6]For example, if 35 reference instances are needed and $l = 10$, the top 4 $|H_{k,i}|$ values are selected for $k = 1, \ldots, 5$, and the top 3 are chosen for $k = 6, \ldots, 10$.

subset spans 6 scenarios for a total of 4285 instances. The list of included and discarded scenarios and sub-scenarios can be found in Appendix A. On this benchmark, the results for 30 LLMs from different families were available at the time we conducted our experiments (see Table 1).

Further, we introduce KindsOfReasoning, a collection of 22 existing datasets, for a total of 37,529 instances. The datasets were selected to cover a wide range of kinds of reasoning (logical, common sense, inductive, deductive, abductive, counterfactual, causal, analogical, spatial and arithmetic reasoning). In particular, we conducted a keyword search in known benchmark repositories (BIG-Bench, Srivastava et al., 2022 and HELM, Liang et al., 2022) and academic search engines for benchmarks about reasoning. Of those we found, we excluded those that require a large amount of commonsense knowledge (such as SocialIQA, Sap et al., 2019), test the dependence of reasoning abilities on context (such as NeuBAROCO, Ozeki et al., 2024) or whose license did not allow derivative works to be distributed (ART, Collier et al., 2022). The final collection contains datasets with different prompting styles, as true reasoning abilities should be robust to these variations. More information is given in Appendix B.

On this dataset, we tested all instruction-tuned models released from OpenAI, from `text-ada-001`[7] to `gpt-4-0125-preview`, for a total of 14 LLMs (see Table 1). The instance-level outputs of all models will be released, in the spirit of Burnell et al. (2023b).

Table 1: LLMs in $\mathcal{L}^{\text{train}}$, $\mathcal{L}^{\text{val}}$ and $\mathcal{L}^{\text{test}}$ for the generic assessor experiments, on the two considered collection of datasets.

|  | KindsOfReasoning | HELM-Lite |
|---|---|---|
| Train | openai/text-ada-001, openai/text-babbage-001, openai/text-curie-001, openai/text-davinci-001, openai/text-davinci-002, openai/gpt-3.5-turbo-0301, openai/gpt-3.5-turbo-0613, openai/gpt-3.5-turbo-1106 | 01-ai/yi-6b, 01-ai/yi-34b, AlephAlpha/luminous-base, AlephAlpha/luminous-supreme, ai21/j2-grande, ai21/j2-jumbo, cohere/command, google/text-bison@001, google/text-unicorn@001, mistralai/mixtral-8x7b-32kseqlen, mistralai/mistral-7b-v0.1, openai/gpt-3.5-turbo-0613, openai/gpt-4-1106-preview, openai/text-davinci-002, openai/text-davinci-003, tiiuae/falcon-7b, writer/palmyra-x-v3, writer/palmyra-x-v2 |
| Validation | openai/text-davinci-003, openai/gpt-3.5-turbo-0125 | tiiuae/falcon-40b, openai/gpt-4-0613, AlephAlpha/luminous-extended, cohere/command-light |
| Test | openai/gpt-4-0125-preview, openai/gpt-4-0314, openai/gpt-4-0613, openai/gpt-4-1106-preview | anthropic/claude-2.1, anthropic/claude-2.0, anthropic/claude-instant-1.2, anthropic/claude-v1.3, meta/llama-2-70b, meta/llama-2-13b, meta/llama-2-7b, meta/llama-65b |

For both of these collections, we shuffle together all datasets and sample a random train, validation, and test splits $\mathcal{D}^{\text{train}}, \mathcal{D}^{\text{val}}$ and $\mathcal{D}^{\text{test}}$ with respective sizes 56%, 14% and 30% of the total number of instances. In Sec. 4.5, we discuss results with OOD splits. Moreover, we identify a split of train, validation, and test LLMs $\mathcal{L}^{\text{train}}, \mathcal{L}^{\text{val}}$ and $\mathcal{L}^{\text{test}}$ for each collection. We make $\mathcal{L}^{\text{test}}$ as different as possible from $\mathcal{L}^{\text{train}}$ and $\mathcal{L}^{\text{val}}$: concretely, we select LLMs from two producers as $\mathcal{L}^{\text{test}}$ for HELM-Lite and all versions of `gpt4` for KindsOfReasoning. In this way, we test the performance of our proposed methodology when the new LLM we want to predict performance for is substantially different from the previously seen ones. The LLM splits are given in Table 1.

## 4.2 METRIC

As a performance metric for the assessors, we use the Area Under the Curve (AUC) which measures how well a binary probabilistic classifier discriminates between the two classes: a classifier assigning non-overlapping probabilities to the two classes achieves the maximum value AUC = 1, while a classifier assigning random values to the two classes achieves AUC = 0.5. As the extreme values of the AUC are insensitive to the class proportion, it can be used to compare results across

---

[7]The older models have been discontinued in January 2024, but we obtained our raw results before that date.

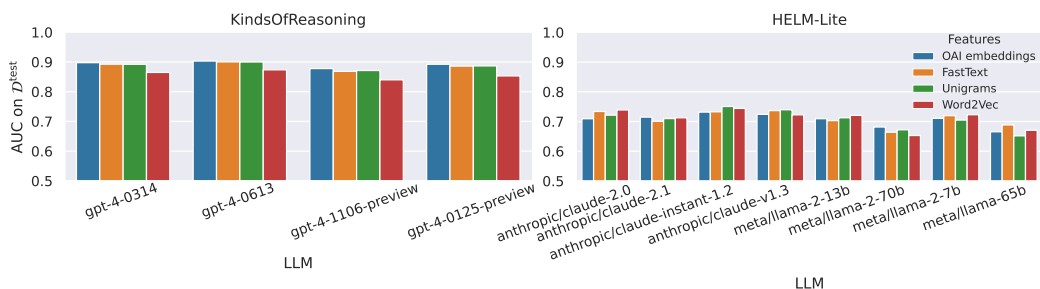

Figure 2: Predictive performance (AUC) of specific assessors for each of the test LLMs $\mathcal{L}^{\text{test}}$ for the two dataset collections, for different prompt features.

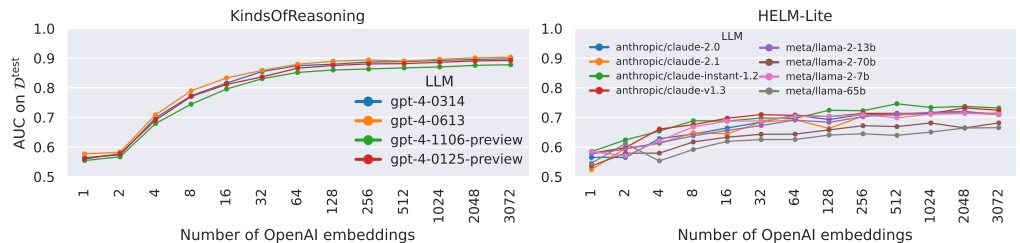

Figure 3: Predictive performance (AUC) of specific assessors for each of the test LLMs $\mathcal{L}^{\text{test}}$ for the two dataset collections, with an increasing number of OpenAI embeddings (endpoint `text-embedding-3-large`).

various scenarios (such as the two dataset collections and different train/validation/test splits). However, the AUC is insensitive to monotonic transformation of the output probabilities, implying that a classifier achieving AUC = 1 can be miscalibrated (such as a classifier assigning probability 0.51 to all positive samples and 0.49 to all negative samples).

### 4.3 WHAT PROMPT FEATURES LEAD TO BETTER PREDICTIVE PERFORMANCE?

We first train an assessor specific to each considered LLM to identify the set of prompt features $f$ that maximizes predictive performance. In particular, we consider the prompt embeddings computed from the OpenAI API (with the `text-embedding-3-large` endpoint OpenAI (2024b)) and those obtained with `Word2vec` (Mikolov et al., 2013) and `FastText` (Bojanowski et al., 2017); the latter two generate a vector for each word in the prompt, which we average to form a vector representing the entire prompt. Lastly, we consider 1-gram vectors, calculated as the frequency of words in a specific prompt, normalized by the frequencies across the entire set of training prompts. For each choice of features and test LLM, we train various base classifiers (logistic regression with $l_2$ and $l_1$ penalty and `xgboost`) on $\mathcal{D}^{\text{train}}$, compute the AUC of each on $\mathcal{D}^{\text{val}}$, pick the one with the highest validation AUC, and report the AUC of that classifier on $\mathcal{D}^{\text{test}}$.

Our results, available in Fig. 2, show that the OpenAI embeddings always perform better for the KindsOfReasoning dataset, while no clear winner emerges for HELM-Lite, where all features lead to similar performance . Therefore, we will use the OpenAI embeddings in all experiments below. Moreover, the OpenAI embeddings obtained from the endpoint `text-embedding-3-large` were trained using Matryoshka Representation Learning (Kusupati et al., 2022), which allows them to be truncated (by removing the final elements of the vector) without the embedding losing its concept-representing properties. As such, we investigate the performance of the specific assessor by truncating the OpenAI embeddings (Fig 3) and found that the performance saturates using 1024 (out of a total of 3072) embeddings; hence, we'll apply this truncation below.

Table 2: The best setup for the generic assessor experiment, selected according to the performance on validation LLMs as discussed in Section 4.4.

| | Instance-intrinsic features | $\mathcal{D}^{\text{ref}}$ selection method | Classifier |
|---|---|---|---|
| *KindsOfReasoning* | Similarity | Random best of 20 | XGBoost |
| *HELM-Lite* | Similarity with interaction | Clustering embeddings | Logistic Regression L1 C=0.1 |

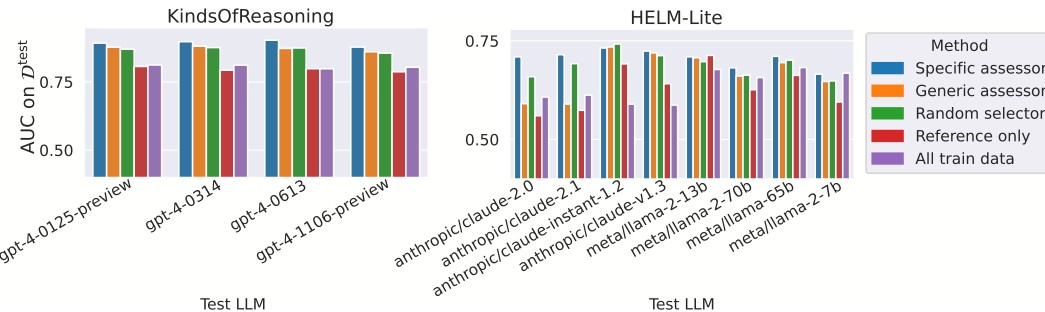

Figure 4: Predictive performance (AUC) of the specific and generic assessor and a few baselines, for the in-distribution experiment on the KindsOfReasoning and HELM-Lite collections of datasets.

## 4.4 GENERIC ASSESSOR PERFORMANCE

Next, we study the predictive performance of the generic assessor. In particular, as instance features $f$ (see Sec. 3.2), we test using the first 1024 elements of the OpenAI embeddings, as well as the cosine similarity of the embeddings between the considered instance and the reference ones, with and without pairwise interaction[8]. To select the reference instances, we test all methods introduced in Sec. 3.2.2. Finally, we test different base classifiers to build the assessors (logistic regression with $l_2$ and $l_1$ penalty and `xgboost`).

For all combinations of instance features, reference dataset selection method and base classifiers, we test our procedure with $\mathcal{D}^{\text{ref}}$ of sizes $30, 100, 300$ and $1000$, for both HELM-Lite and KindsOfReasoning. We found that the validation AUC of the classifier approximately saturated for 100 reference samples (see Appendix C). As such, we use this that size of $\mathcal{D}^{\text{ref}}$ below.

Next, we evaluate the AUC of each combination of classifier, selection of $\mathcal{D}^{\text{ref}}$ and instance features $f$ on $\mathcal{D}^{\text{val}}$ for each LLM in $\mathcal{L}^{\text{val}}$. We then compute the win rate of each combination for each validation LLM and pick the combination with the highest average win rate over $\mathcal{L}^{\text{val}}$ (a simple average over $\mathcal{L}^{\text{val}}$ would be impacted by the intrinsic different predictability of the different LLMs, which change the maximally achievable AUC). The winning combination is reported in Table 2. Interestingly, for the KindsOfReasoning collection, the randomly sampled $\mathcal{D}^{\text{ref}}$ performs better than those determined according to the advanced criteria in Section 3.2.1. While surprising at first, other works (Ye et al., 2023; Wang et al., 2023; Polo et al., 2024; Kipnis et al., 2024) had found that benchmarks can be reduced by random sampling for multiple purposes. Next, we compare the performance of the winning combination on $\mathcal{D}^{\text{test}}$, alongside the specific assessor (which relies on the test LLM results on $\mathcal{D}^{\text{train}}$ and $\mathcal{D}^{\text{val}}$) and three baselines:

- "Random selector", a generic assessor where $\mathcal{D}^{\text{ref}}$ is randomly selected.

- "Reference only", where, for each $\mathcal{L}^{\text{test}}$, we train an assessor only using the prompt features and the performance of the elements of $\mathcal{D}^{\text{ref}}$ (thus, ignoring the previous LLMs).

- "All train data", obtained by fitting a single assessor on the pooled performance results of all LLMs in $\mathcal{L}^{\text{train}}$ on $\mathcal{D}^{\text{train}}$ only using the intrinsic features $f(p_i)$ (effectively considering all LLMs as a single LLM and ignoring the new LLM's performance on $\mathcal{D}^{\text{ref}}$).

---

[8]Notice how the set of reference instances is fixed for all LLMs in $\mathcal{L}^{\text{test}}$, so the similarities are independent of the considered test LLM.

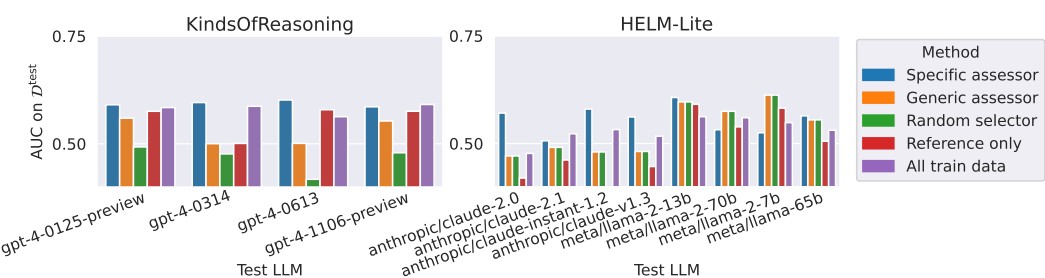

Figure 5: Predictive performance (AUC) of the specific and generic assessor and a few baselines, for a chosen OOD split of the KindsOfReasoning and HELM-Lite collections of datasets.

The results are reported in Fig. 4. The specific assessor always outperforms our generic assessor, as expected from the former having access to more information about the test LLM; however, the performance gap is generally small. Further, the generic assessor almost always outperforms or performs comparably with the "all train data" and "reference only" baselines, indicating that combining the information on previous LLMs and the evaluation results of the test LLM on $\mathcal{D}^{\text{ref}}$ generally performs better than relying only on either one. Moreover, the generic assessor and the "random selector" baseline often perform comparably and there are a few cases where either one prevails, in roughly equal frequency; in particular, two LLMs for HELM-Lite show much better performance with the random selector. This indicates that the generic assessor is not sensitive to the specific selection of $\mathcal{D}^{\text{ref}}$. Notice how, on validation data, the selected combination of selector, features, and classifier for the generic assessor is always better than the random selector baseline, as the possible choices for the latter are a subset of those for the former; however, our Figure 4 shows how, at least in a few cases, it is possible that the random selector performs better on test data.

### 4.5 OUT-OF-DISTRIBUTION STUDY

We repeat the full set of experiments by considering multiple out-of-distribution (OOD) splits for the HELM-Lite and KindsOfReasoning collections, where we keep one set of datasets as $\mathcal{D}^{\text{test}}$ (according to some criteria), and obtain $\mathcal{D}^{\text{train}}$ and $\mathcal{D}^{\text{val}}$ by randomly shuffling the remaining ones. The complete description and results are available in Appendix D; here, we only report results on a chosen OOD split for the experiment comparing the generic assessor with the baselines and the specific assessor. From the results, in Fig. 5, it can be seen how the overall predictive power is decreased and there is no clear ranking of the various methods as was found in the in-distribution experiments (Sec. 4). The results in Appendix further confirm this finding.

## 5 CONCLUSION

We proposed a novel framework for predicting the performance of a new LLM on individual task instances by leveraging the evaluation results of previously tested LLMs. Our approach minimises the number of evaluations required for a new LLM by introducing a *generic assessor* combining instance-specific features with LLM-specific ones derived from performance on a small set of reference instances. While we focus on LLMs, our methodology can be seamlessly applied to predict the performance of other AI systems, by using suitable system-specific and instance-specific features. Similarly, our approach can also be extended to non-binary correctness metrics, the investigation of which we leave to future work.

Our empirical studies on the HELM-Lite and KindsOfReasoning dataset collections showed how the generic assessor performs only slightly worse than the specific one in distribution, while outperforming simpler baselines. Moreover, we found that the generic assessor is mostly unsensitive to the specific set of reference instances used. Finally, out of distribution, the predictive performance decreases drastically for all methods, which raises awareness of the low inner predictability of LLMs. To foster research in making LLMs more predictable (Zhou et al., 2023), we release the instance-level results of all instruction-finetuned GPT3 and GPT4 models until `gpt4-0125-preview` on KindsOfReasoning.

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

## A  INFORMATION ON THE EXCLUDED SCENARIOS FROM HELM-LITE

As mentioned in the main text, we discard some scenarios and subscenarios from HELM-Lite as either the performance metric was non-binary or because the available results used a different number of few-shot prompts for different LLMs. In particular, we discard the following:

- LegalBench:
    - corporate lobbying - incoherent number of few-shots across LLMs
- MATH:
    - algebra - incoherent number of few-shots across LLMs
    - geometry - incoherent number of few-shots across LLMs
    - intermediate algebra - incoherent number of few-shots across LLMs
- NarrativeQA: non-binary metric (f1 score)
- NaturalQuestions: non-binary metric (f1 score)
- WMT 2014: non-binary metric (BLEU score)

As such, the subset of HELM-Lite that we consider throughout our experiments is made up of the following scenarios and subscenarios:

- commonsense
- GSM8K
- MedQA
- LegalBench:
    - abercrombie
    - function of decision section
    - proa
    - international citizenship questions
- MATH:
    - counting and probability
    - number theory
    - prealgebra
    - precalculus
- MMLU:
    - abstract algebra
    - college chemistry
    - computer security
    - econometrics
    - US foreign policy

## B  THE KINDSOFREASONING COLLECTION

Table 3 shows detailed information on the datasets included in the KindsOfReasoning collection. For some datasets, we only kept a smaller number of instances than the one available, to reduce the cost of evaluating a model on the full benchmark. We do not do this for the "Arithmetic" dataset as each of the prompt of that dataset is short, and hence the cost of evaluating it is small (besides, we use Arithmetic as the test data for one of our chosen splits, and subsampling it would have made the test data too small).

Table 3: Datasets used in building the KindsOfReasoning collection. See Appendix B for information on the column meanings.

| Task name | Reasoning type | Used in | Task Type | Used split | N samples | N samples used | Notes | Source used |
|---|---|---|---|---|---|---|---|---|
| formal fallacies syllogisms negation (Srivastava et al., 2022) | Logical reasoning | BIG-Bench | Valid/invalid | - | 14200 | 1000 | - | BIG-Bench |
| logical_args (Srivastava et al., 2022) | Logical reasoning common sense | BIG-Bench | MC (5) | - | 32 | 32 | - | BIG-Bench |
| babi_task_16 (Srivastava et al., 2022) | inductive reasoning | LogiGLUE | 1-word answer | test | 5000 | 1000 | - | BIG-Bench |
| LogiQA 2.0 (Liu et al., 2023) | deductive reasoning | LogiGLUE GLoRE | MC (4) | validation | 1569 | 1569 | [9] | OpenAI `evals` library |
| wanli (Liu et al., 2022) | deductive reasoning | LogiGLUE | NLI | test | 5000 | 1000 | Slightly modified the prefix | LogiGLUE |
| alpha_nli (Bhagavatula et al., 2019) | abductive | CALM-bench LogiGLUE | MC (2) | test | 1432 | 1000 | Changed from NLI to MC format | LogiGLUE |
| reclor (Yu et al., 2020) | abductive, inductive, deductive reasoning | LogiGLUE GLoRE | MC (4 options) | test | 500 | 500 | [10] | OpenAI `evals` library |
| crass_ai (Srivastava et al., 2022) | Counterfactual reasoning | BIG-Bench | MC (5 options) | - | 44 | 44 | - | BIG-Bench |
| cause and effect (Srivastava et al., 2022) | Causal reasoning | BIG-Bench | MC (2) | - | 102 | 102 | Over 2 different formats | BIG-Bench |
| fantasy reasoning (Srivastava et al., 2022) | Causal reasoning | BIG-Bench | Yes/No | - | 201 | 201 | - | BIG-Bench |
| goal step inference (Srivastava et al., 2022) | Causal reasoning | BIG-Bench | MC (4) | - | 7053 | 3000 | Over 3 subtasks | BIG-Bench |
| Copa (Gordon et al., 2011) | Causal reasoning, world knowledge | CALM-bench | MC (2) | test | 500 | 500 | - | Original source |
| Cosmos_qa (Huang et al., 2019) | Causal reasoning, world knowledge | CALM-bench | MC (4) | validation | 2985 | 2985 | use validation set as the test set does not have labels. | HuggingFace |
| ropes(Lin et al., 2019) | Causal reasoning, world knowledge | CALM-bench | Completion | validation | 1688 | 1688 | use validation set as the test set does not have labels. | HuggingFace |
| Anli (Nie et al., 2019) | Causal reasoning, world knowledge | LogiGLUE | NLI | test | 3200 | 3200 | Merged the 3 "rounds" (levels of difficulty) together | Original source |
| Emoji_movie (Srivastava et al., 2022) | analogical reasoning, world knowledge | BIG-Bench | MC (5) | - | 100 | 100 | - | BIG-Bench |
| abstract narrative understanding (Srivastava et al., 2022) | analogical reasoning | BIG-Bench | MC (10 and 100) | - | 2000 | 2000 | Over 2 subtasks (9 and 99 distractors; I discarded the one with 4 distractors) | BIG-Bench |
| odd one out (Srivastava et al., 2022) | analogical reasoning | BIG-Bench | MC (variable number) | - | 86 | 86 | - | BIG-Bench |
| metaphor understanding (Srivastava et al., 2022) | analogical reasoning | BIG-Bench | True/False | - | 680 | 680 | - | BIG-Bench |
| geometric shapes (Srivastava et al., 2022) | Spatial reasoning | BIG-Bench | MC (10) | - | 360 | 360 | - | BIG-Bench |
| Space_nli (Abzianidze et al., 2023) | Spatial reasoning | - | NLI | - | 1604 | 1604 | - | Original source |
| Arithmetic (Srivastava et al., 2022) | Arithmetic ability | BIG-Bench | Completion | - | 15023 | 15023 | Over 20 subtasks | BIG-Bench |

Most of the datasets included in this collection are present in one (or more) of BIG-Bench (Srivastava et al., 2022), LogiGLUE (Luo et al., 2023), CALM-bench (Dalal et al., 2023) and GLoRE (Teng et al., 2023). However, as mentioned in the main text (Sec. 2), our collection covers more kinds of reasoning. The dataset and the instance-level results of all instruct-GPT models from OpenAI (from `text-ada-001` to `gpt4-0125-preview` will be released at `anonymised`).

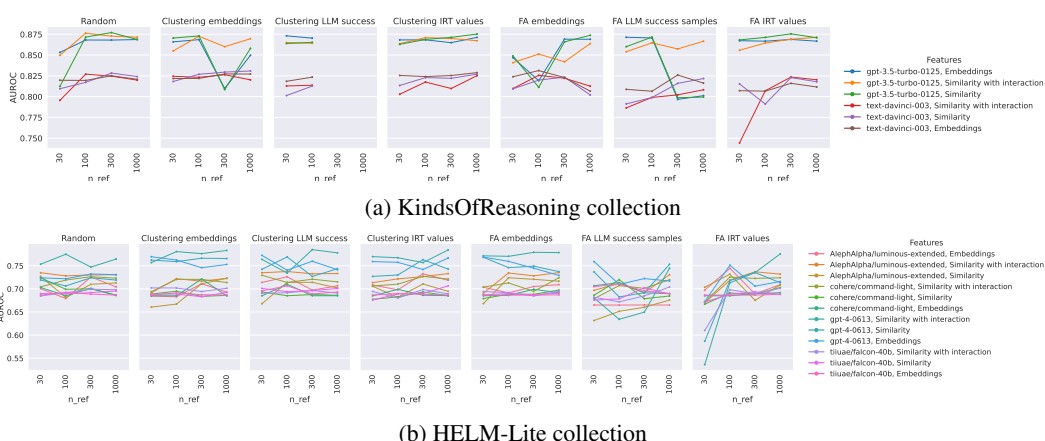

(a) KindsOfReasoning collection

(b) HELM-Lite collection

Figure 6: AUC with increasing number of reference instances on the validation data split, for the various validation LLMs, reference dataset selection methods and considered instance features. The "Clustering LLM successes" for the KindsOfReasoning collection failed to converge for $\mathcal{D}^{\text{ref}}$ of size 300 and 1000.

## C  IMPACT OF THE NUMBER OF REFERENCE POINTS

Figure 6 shows the performance (AUC) of the generic assessor for different values of the number of reference points selected, reference dataset selection methods and instance features, for the validation LLMs ($\mathcal{L}^{\text{val}}$) on the validation split $\mathcal{D}^{\text{val}}$ of the KindsOfReasoning (top panels) and HELM-Lite (bottom panels) collection respectively. For each value of the number of reference points and each reference dataset selection method, multiple classifiers were trained, and the one with the highest AUC is reported. Broadly, it can be seen as the performance on $\mathcal{D}^{\text{val}}$ roughly peaks at around 100 reference instances (although a few cases are roughly constant and some others show a drop for higher number of reference instances). Notice that the "Clustering LLM successes" for the KindsOfReasoning collection failed to converge for $\mathcal{D}^{\text{ref}}$ of size 300 and 1000.

## D  OUT OF DISTRIBUTION STUDY

We repeat all experiments discussed in the main text (Sec. 4) considering different choices for the train, validation, and test splits $\mathcal{D}^{\text{train}}, \mathcal{D}^{\text{val}}$ and $\mathcal{D}^{\text{test}}$ for both dataset collections. The main text reported results with a random split, where the various splits are sampled by shuffling together all instances of all datasets. In addition, we consider multiple out-of-distribution (OOD) splits, where we keep one set of datasets as $\mathcal{D}^{\text{test}}$ (according to some criteria), and $\mathcal{D}^{\text{train}}$ and $\mathcal{D}^{\text{val}}$ are obtained from randomly shuffling the other ones. In this way, the data used to train and select the best assessor (both in the generic and specific setup) have the same distribution, which is however different from the data where the selected assessor will be evaluated on. Details on the various splits are given in Table 4. Section 4.5 in the main text reported results using the second OOD splits for both dataset collections.

First, as done in Sec. 4.3 for the in-distribution case, we compute the predictive performance of specific assessors built on different features intrinsic to the prompt, for the different data splits of the KindsOfReasoning and HELM-Lite collections respectively. Results are reported in Figures 7 and 8; in particular, for each figure, the top panel shows performance on $\mathcal{D}^{\text{val}}$, while the latter shows performance on $\mathcal{D}^{\text{test}}$, for the classifier selected according to its best performance on $\mathcal{D}^{\text{val}}$. On the validation data, the performance of the OpenAI embeddings is generally higher and, as such, the experiments reported in the main text are with this choice of embeddings. However, the performance

---

[9]I use the multiple-choice version rather than the NLI one; moreover, the source I used shuffled the order of options and replaced the correct option with "none is correct", so the model should always select that.

[10]The source I used shuffled the order of options and replaced the correct option with "none is correct", so the model should always select that.

Table 4: Size of $\mathcal{D}^{\text{train}}$, $\mathcal{D}^{\text{val}}$ and $\mathcal{D}^{\text{test}}$ for the different splits for the KindsOfReasoning and HELM-Lite collections, together with the criteria for which datasets to include in the test split ($\mathcal{D}^{\text{train}}$ and $\mathcal{D}^{\text{val}}$ are randomly obtained from those not included in $\mathcal{D}^{\text{test}}$).

| | Train size | Validation size | Test size | Test set composition |
|---|---|---|---|---|
| | | *KindsOfReasoning* | | |
| In-distribution | 21016 | 5254 | 11259 | Random |
| OOD 1 | 18069 | 4517 | 14943 | arithmetic |
| OOD 2 | 20705 | 5176 | 11648 | causal |
| OOD 3 | 21273 | 5318 | 10938 | logical, deductive, inductive, spatial, abductive, counterfactual, and analogical reasoning |
| OOD 4 | 23238 | 5810 | 8481 | world knowledge, common sense |
| | | *HELM-Lite* | | |
| In-distribution | 2400 | 600 | 1285 | Random |
| OOD 1 | 2378 | 595 | 1312 | Math, GSM, MMU abstract algebra |
| OOD 2 | 2182 | 546 | 1557 | Legalbench |
| OOD 3 | 2295 | 574 | 1416 | Commonsense, Med QA, MMLU (except abstract algebra) |

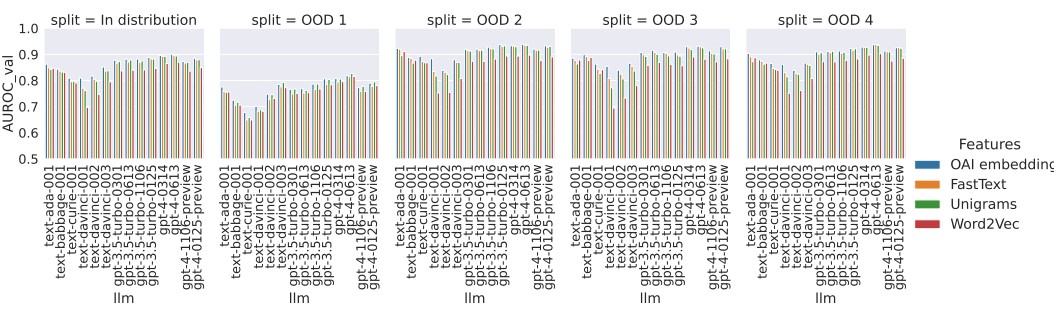

(a) AUC with different choices of instance-intrinsic features on $\mathcal{D}^{\text{val}}$.

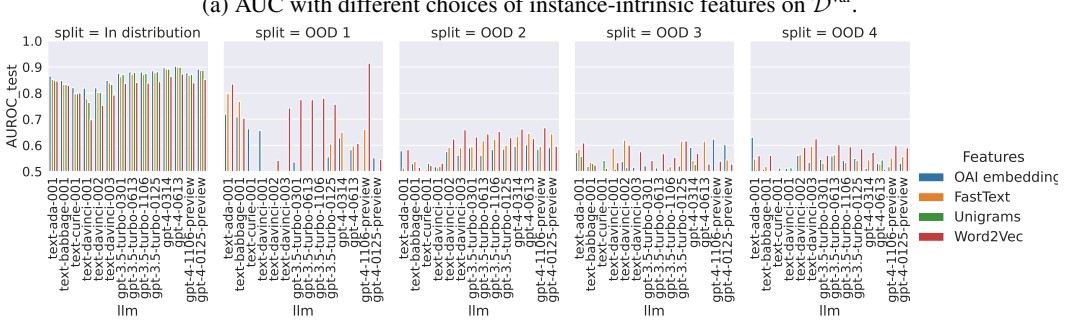

(b) AUC with different choices of instance-intrinsic features on $\mathcal{D}^{\text{test}}$.

Figure 7: AUC with different choices of instance-intrinsic features (OpenAI embeddings, Word2Vec, FastText and 1-gram), for different splits on KindsOfReasoning. For each split and feature, various classifiers were trained on $\mathcal{D}^{\text{train}}$ and the best according to its performance on $\mathcal{D}^{\text{val}}$ was selected; the panels report the performance of the latter on $\mathcal{D}^{\text{val}}$ and $\mathcal{D}^{\text{test}}$.

on $\mathcal{D}^{\text{test}}$ for the OOD splits show a mixed picture, with the OpenAI embeddings often performing worse than simpler ones (such as Word2Vec) and with generally lower performance.

Next, we compute the performance of the specific assessor using the OpenAI embeddings truncated at different vector sizes, for different data splits of the KindsOfReasoning and HELM-Lite collec-

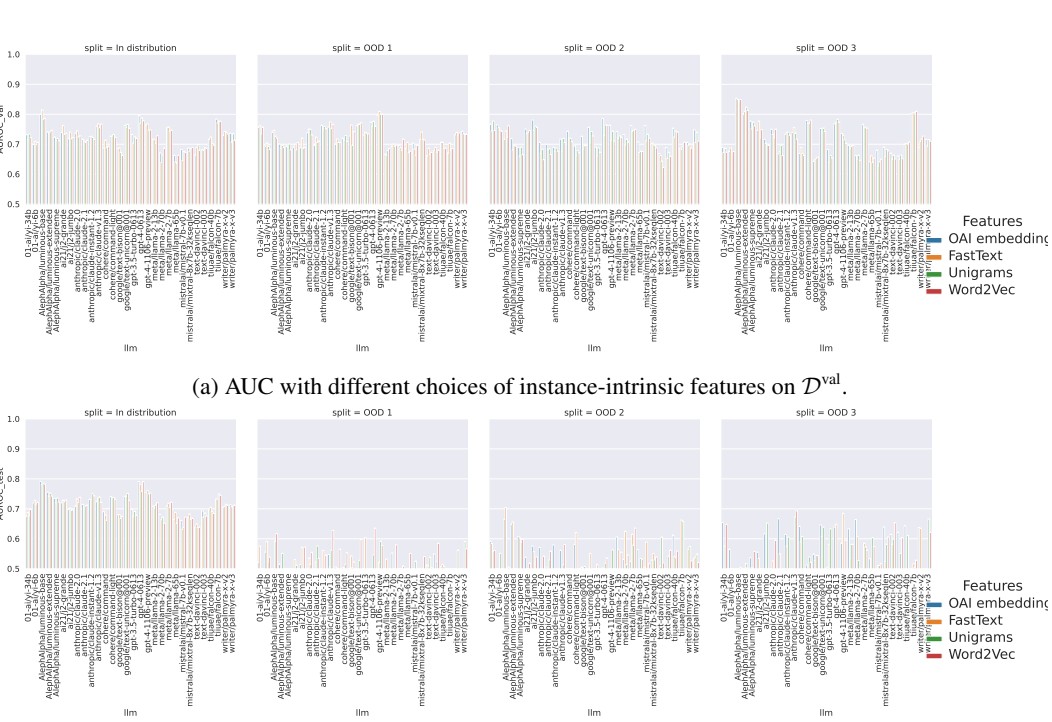

(a) AUC with different choices of instance-intrinsic features on $\mathcal{D}^{\mathrm{val}}$.

(b) AUC with different choices of instance-intrinsic features on $\mathcal{D}^{\mathrm{test}}$.

Figure 8: AUC with different choices of instance-intrinsic features (OpenAI embeddings, Word2Vec, FastText and 1-gram), for different splits on HELM-Lite. For each split and feature, various classifiers were trained on $\mathcal{D}^{\mathrm{train}}$ and the best according to its performance on $\mathcal{D}^{\mathrm{val}}$ was selected; the panels report the performance of the latter on $\mathcal{D}^{\mathrm{val}}$ and $\mathcal{D}^{\mathrm{test}}$.

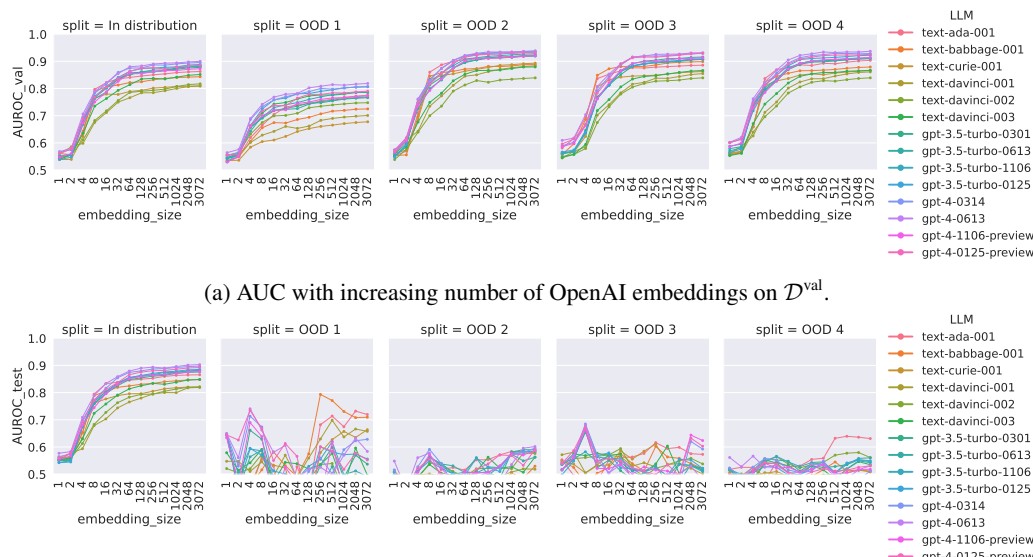

(a) AUC with increasing number of OpenAI embeddings on $\mathcal{D}^{\text{val}}$.

(b) AUC with increasing number of OpenAI embeddings on $\mathcal{D}^{\text{test}}$.

Figure 9: AUC with increasing number of OpenAI embeddings for specific assessors trained on increasing number of OpenAI embeddings, for different splits on KindsOfReasoning. For each split and number of embeddings, various classifiers were trained on $\mathcal{D}^{\text{train}}$ and the best according to its performance on $\mathcal{D}^{\text{val}}$ was selected; the panels report the performance of the latter on $\mathcal{D}^{\text{val}}$ and $\mathcal{D}^{\text{test}}$.

tions respectively. The results are in Figures 9 and 10; in particular, for each figure, the top panel shows performance on $\mathcal{D}^{\text{val}}$, while the latter shows performance on $\mathcal{D}^{\text{test}}$, for the classifier selected according to its best performance on $\mathcal{D}^{\text{val}}$. The performance on $\mathcal{D}^{\text{val}}$ (and $\mathcal{D}^{\text{test}}$ for the in-distribution split) plateaus when the truncation size reaches 1024 and, as such, all the results reported in the main text are with that truncation size. On $\mathcal{D}^{\text{test}}$ for the various OOD splits, the performance does not follow a smooth curve, but still seems to peak more often around a truncation size of 1024.

We then move on to considering the generic assessor setup, and we select the best combination of selector method, instance features and base classifiers as done for the in-distribution study in Sec. 4.4. The winning combination for each data split is reported in Table 5. Interestingly, for multiple data splits, the randomly sampled $\mathcal{D}^{\text{ref}}$ performs better than those determined according to the advanced criteria in Section 3.2.1. While surprising at first, other works (Ye et al., 2023; Wang et al., 2023; Polo et al., 2024; Kipnis et al., 2024) had found that benchmarks can be reduced by random sampling for multiple purposes. In terms of classifier, instead, XGBoost generally performs better. Finally, using similarity between the embeddings of the reference instances and those of the considered instance more frequently performs better than directly using the latter as $f(p)$.

Finally, Figure 11 reports the performance results of the best combination for the generic assessor and the specific assessor, alongside the baselines introduced in Sec. 4.4. From those results, several considerations can be made. First, notice how the predictive performance generally degrades out of distribution with respect to the in-distribution (random) split. For some out-of-distribution splits, some predictive power remains (recall that AUC = 0.5 corresponds to random guess) but, on other splits, even the specific assessor performs poorly, despite relying on evaluation results of the test LLMs on the whole train and validation data splits. This indicates that the considered intrinsic features of the prompt (the OpenAI embeddings) do not reliably capture a general performance pattern. While, in principle, more informative features could be used, it is also possible that there is an inner limit to the out-of-distribution predictability of the current generation of LLMs, due to their stochastic nature.

Moreover, the specific assessor always outperforms our generic assessor in distribution and does so frequently out of distribution, as expected from the former having access to more information about the test LLM; however, the performance gap is generally small. In distribution, further, the generic assessor almost always outperforms or performs comparably with the "all train data" and

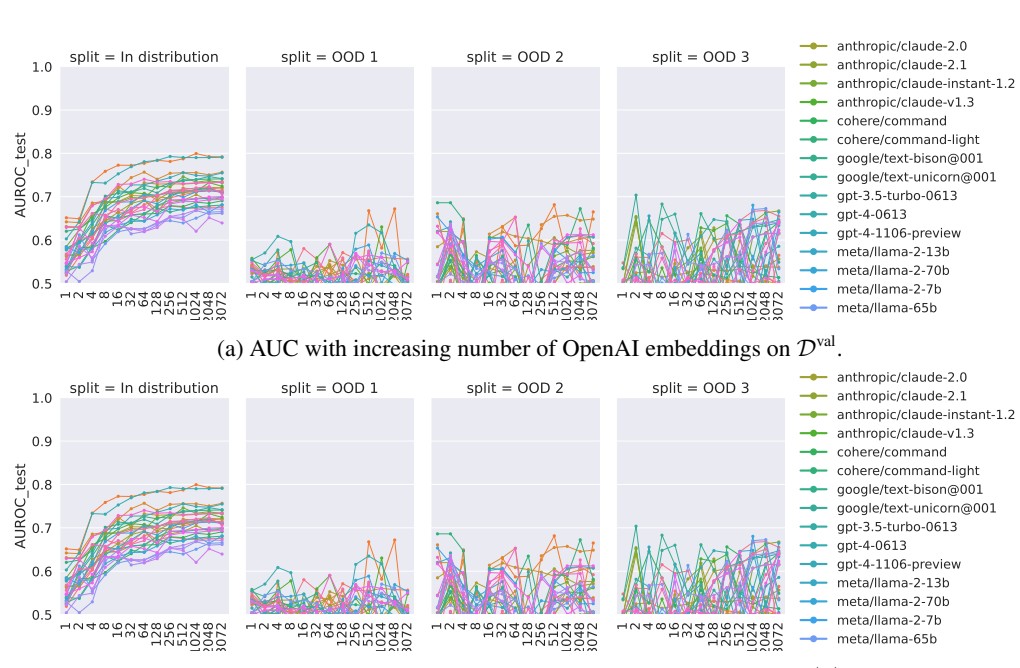

(a) AUC with increasing number of OpenAI embeddings on $\mathcal{D}^{\text{val}}$.

(b) AUC with increasing number of OpenAI embeddings on $\mathcal{D}^{\text{test}}$.

Figure 10: AUC with increasing number of OpenAI embeddings for specific assessors trained on increasing number of OpenAI embeddings, for different splits on HELM-Lite. For each split and number of embeddings, various classifiers were trained on $\mathcal{D}^{\text{train}}$ and the best according to its performance on $\mathcal{D}^{\text{val}}$ was selected; the panels report the performance of the latter on $\mathcal{D}^{\text{val}}$ and $\mathcal{D}^{\text{test}}$.

Table 5: The best combination of instance-intrinsic features, selector and classifier for each data split in the two considered dataset collections, selected according to the performance on validation LLMs as discussed in Section 4.4. In the "instance-intrinsic features" column, "embeddings" refers to using the OpenAI embeddings of the considered instance as $f(p_i)$, while "similarity" refers to using the cosine similarity between the OpenAI embeddings of the reference instances and that of the considered instance; further, "similarity with interaction" explicitly adds features obtained as the pairwise produce of each similarity with its corresponding success (notice that this is superfluous for XGBoost, which can natively leverage interactions between features).

|  | Instance-intrinsic features | Selector | Classifier |
|---|---|---|---|
| | *KindsOfReasoning* | | |
| In-distribution | Similarity | Random best of 20 | XGBoost |
| OOD 1 | Similarity | Factor analysis embeddings | XGBoost |
| OOD 2 | Similarity with interaction | Clustering IRT values | XGBoost |
| OOD 3 | Embeddings | Random | XGBoost |
| OOD 4 | Similarity | Random | XGBoost |
| | *HELM-Lite* | | |
| In-distribution | Similarity with interaction | Clustering embeddings | Logistic Regression L1 C=0.1 |
| OOD 1 | Embeddings | Clustering LLM success | XGBoost |
| OOD 2 | Similarity with interaction | Random | Logistic Regression L1 C=1 |
| OOD 3 | Similarity with interaction | Clustering LLM success | Logistic Regression L1 C=1 |

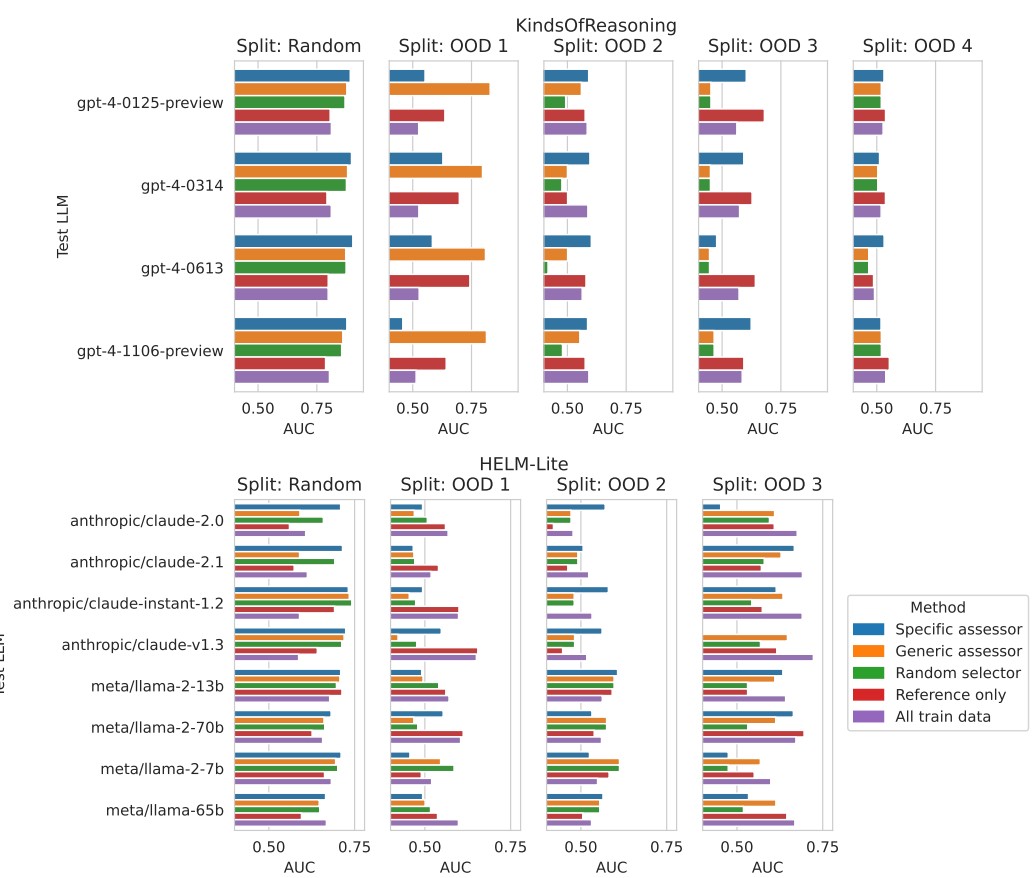

Figure 11: Predictive performance (AUC) of the specific and generic assessor and a few baselines, for different splits of the KindsOfReasoning and HELM-Lite collections of datasets. Some combinations (for instance, the random selector on split 1 of KindsOfReasoning achieve AUC lower than the lower bound of the panels (0.4) and are hence hidden in the graph.

"reference only" baselines, indicating that combining the information on previous LLMs and the evaluation results of the test LLM on $\mathcal{D}^{\text{ref}}$ generally performs better than relying only on either one. For some OOD splits (OOD 2 and 3 for KindsOfReasoning and OOD1 and 3 for HELM-Lite), instead, either or both of these baselines perform better than the generic assessor, indicating how the generic assessor likely overfits to the training distribution; however, in most of those cases, the predictive performance is quite low for all methods (except for split 3 in HELM-Lite).

If we instead compare the generic assessor with the "random selector" baseline (which is identical to the generic assessor but with a random $\mathcal{D}^{\text{ref}}$), we see how the two often perform comparably and there are a few cases where either one prevails, in roughly equal frequency. This indicates that the generic assessor is not sensitive to the specific selection of $\mathcal{D}^{\text{ref}}$ (an indication for this could also be seen in Table 5, where there is no coherent best selector and where a few times the "random" subset was selected as best). Notice how, on validation data, the selected combination of selector, features, and classifier for the generic assessor is always better than the random selector baseline, as the possible choices for the latter are a subset of those for the former; however, our Figure 11 shows how, at least in a few cases, it is possible that the random selector performs better on test data.

In a similar manner, the "reference only" baseline is identical to a "specific assessor" trained on a subset of $\mathcal{D}^{\text{train}}$, but with the selection of the best classifier being carried out on the validation LLMs, instead of using the results of the considered LLM on $\mathcal{D}^{\text{val}}$. Still, the specific assessor always performs better than "reference only" in-distribution, while the latter sometimes overtakes the former out-of-distribution, indicating that the specific assessor overfits the training distribution due to the larger number of training points or due to the classifier selection being performed using the test LLM.

# E  CONTROL FOR NUMBER OF TRAINING SAMPLES IN THE KINDSOFREASONING COLLECTION

Figure 12 shows the difference between the AUC of a specific assessor trained on the full $\mathcal{D}^{\text{train}}$ and one trained on a random subsample of $\mathcal{D}^{\text{train}}$ of size 3000, for different choices of the random split for the KindsOfReasoning collection. The difference is small on $\mathcal{D}^{\text{val}}$ (notice the $y$ scale of the graphs) and generally small for $\mathcal{D}^{\text{test}}$ for all data splits, except for OOD 1, which reaches higher absolute values on both sides of 0.

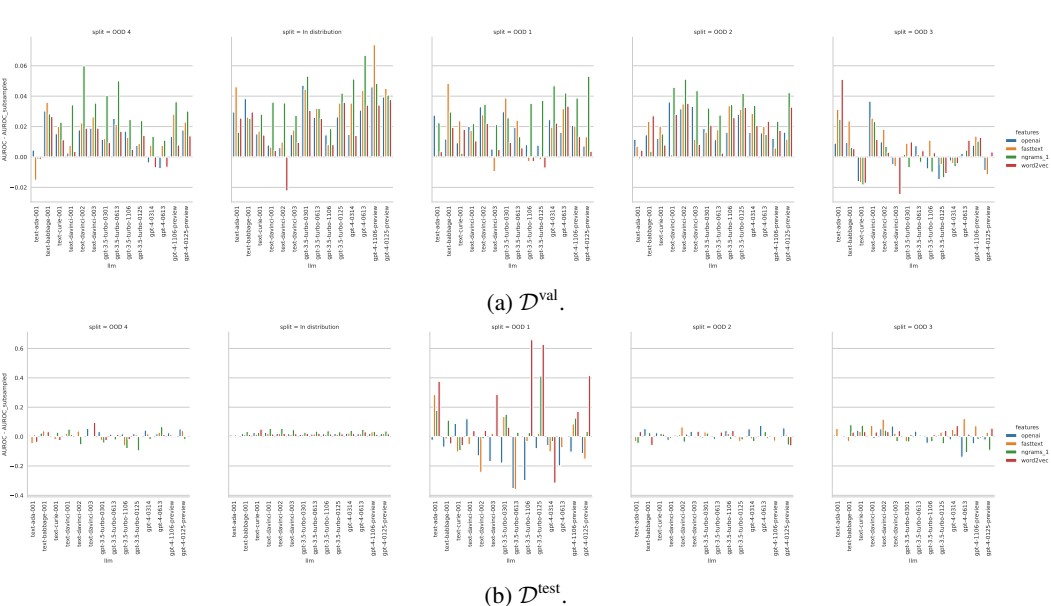

(a) $\mathcal{D}^{\text{val}}$.

(b) $\mathcal{D}^{\text{test}}$.

Figure 12: Difference between the AUC of a specific assessor trained on the full $\mathcal{D}^{\text{train}}$ and one trained on a random subsample of $\mathcal{D}^{\text{train}}$ of size 3000, for different choices of the random split for the KindsOfReasoning collection. Positive values indicate better performance of the specific assessor trained on the full $\mathcal{D}^{\text{train}}$, and viceversa. For each split and feature, various classifiers were trained on $\mathcal{D}^{\text{train}}$ and the best according to its performance on $\mathcal{D}^{\text{val}}$ was selected; the panels report the difference in performance of the latter on $\mathcal{D}^{\text{val}}$ and $\mathcal{D}^{\text{test}}$.

