# OpenReview forum: "100 instances is all you need: predicting LLM success by testing on a few instances"
_ICLR.cc/2025/Conference — Submitted to ICLR 2025_

### Official Review · Reviewer_XEPY · 2024-10-29

**Soundness:** 2
**Presentation:** 3
**Contribution:** 1
**Rating:** 3
**Confidence:** 4

**Summary:**

The authors conduct new empirical studies of generic assessor models, which are trained to predict the performance of any LLM on an instance by using the LLM’s performance on a small set of reference instances and the features of the considered instance, making use of existing evaluation results to extract the representative instances and train the assessor.

The authors find that a few instances (around 100) are enough to train assessors with predictive power comparable to the LLM-specific assessors trained on the complete set of several thousand instances. Interestingly, randomly selecting the reference instances performs comparably to the advanced selection methods.

The authors identify a sharp drop in predictive power of the generic and specific assessors in out-of-distribution
scenarios, suggesting that the inherent predictability of LLMs is low.

**Strengths:**

1. The authors introduce a new metadataset, KindsOfReasoning, which may be useful for future work.
2. The sharp drop in predictive power of the generic and specific assessors in out-of-distribution scenarios should give evidence for future works as to the limited utility of assessors for complex foundation models.
3. The related works section is reasonably comprehensive.

**Weaknesses:**

1. The authors take it as a given that their choice of subject is high impact enough for this venue, and novel enough to require no experimental baselines, only ablation experiments on the construction of the assessor model. However, the evidence supporting this assumption is dubious.

Circa lines 39-40 (the line numbers are not aligned with the text lines), the authors cite 5 works developing assessors, but three of the papers the authors cite describe topics other than assessor models. https://arxiv.org/abs/2309.15789 describes learning a "router" model over tasks for LLM selection; optimally pairing tasks with a fixed pool of LLMs. In other words, it is not an assessor because it does not meet the requirements (it does not predict at an instance level). https://arxiv.org/abs/2107.11277 is a survey on ML with a reject option, a related but distinct topic. https://ieeexplore.ieee.org/abstract/document/10650424 is a methods paper on ML with a reject option. The authors acknowledge as much in Sec. 2.1, where they argue that instance-level prediction of success unifies these works. But then why are they cited as continuing a line of work on assessor models? This is, at best, confusing.

Of those works that do discuss assessor models, https://ojs.aaai.org/index.php/AAAI/article/view/21487 appears to be the first and claims to have originated the topic. The paper is 4 pages long, with minimal experiments. Essentially, it is a position paper, arguing that such models might have value as a conceptual class. https://ceur-ws.org/Vol-3169/paper4.pdf, from many of the same authors, follows it up with a more extended investigation.

In Table 1 of https://ojs.aaai.org/index.php/AAAI/article/view/21487, the authors cite 5 desiderata for assessor models:

**Anticipative**. It is essential that an assessor is able to predict performance before a system is dispatched, even in areas it has never been used.

**Standalone**. An assessor must work independently from the original system, not requiring access to the system or its outputs.

**Granular**. An assessor must predict at instance granularity, and reflect that some situations are easier than others.

**Behavioural**. An assessor must learn representations of the emergent behaviour of the system, without access to system internals (black box approach).

**Distributional**. An assessor’s predictive power will come from populations of (related) systems, but also aggregating estimates conditioned to distributions.

There are robust and extensive areas of study, such as reward models in RL and uncertainty quantification in ML, that fulfill many of these criteria, as well as a half-dozen other overlapping metrics and fields described in https://ojs.aaai.org/index.php/AAAI/article/view/21487. On top of all of that, the authors of this work devote nearly 1.5 pages to related work, much of which closely overlaps with assessor models.

Despite having perhaps hundreds of potential strong baseline methods to compare against, the authors compare their work only to their own constructed assessor baselines, more akin to ablation studies. And even so, many of their findings are negative.

2. KindsOfReasoning is a metadataset; thus, its value as a novel contribution is limited.

**Questions:**

Why is it necessary to establish assessor models as a distinct field of inquiry, given the many related fields which largely overlap with them?

Why do the authors not compare their method to the many existing baseline methods for predicting the performance of models?

---

> ### Author Response · Authors · 2024-11-16
> **Response to reviewer's comments**
>
> We thank the reviewer for their comments and suggestions. We discuss the major points below. We also updated our manuscript following some of the suggestions by the reviewer.
>
> - **Comparison with other baselines:** the main critique by the reviewer is that, given the several fields related to assessors, our manuscript should include comparisons with baseline approaches drawn from these fields. However, we are unaware of any related methods that would be applicable in our case and with the assumptions we consider. The reviewer themselves point out how many approaches satisfy _some_ of the desiderata introduced in [1] for assessor models, but, to the best of our knowledge, no previous method satisfies _all_ of them and is thus applicable to our considered scenario. For instance, uncertainty quantification could be used to extract a measure of confidence from the LLMs themselves, but that would require inputting the considered prompt to the LLM -- in contrast, we are interested in anticipative and standalone methods. We are instead unsure as to how RL reward models could be of use in our scenario, as they are usually trained using model outputs, and would therefore learn a verification mechanism, rather than predict model performance a priori.  Could the reviewer please suggest concrete examples of baselines we could compare to?
>
>
> - **References in the first paragraph:** the reviewer is correct in saying that [1] and [3]  do not develop assessors, despite being closely related works. On the other hand, [4] develops an approach ``to detect instances in regions of data where the base model has shown poor predictive performance during its evaluation'', which corresponds to what an assessor does, a terminology that is also present in that paper. Therefore, we feel referring to that work in the first paragraph is motivated.
>
>
> - **Comments on assessors as a separate field of inquiry:** The reviewer dedicates a large part of the review to summarising previous works on assessor models and pointing out that multiple related fields exist. They then ask why it is necessary to establish assessor models as a separate field of inquiry. We respond that our work does not aim to do so; indeed, the author refers to [1], which established this direction of investigation. Our work aims to introduce a novel method building on previous works in this line of research. We agree with the reviewer that several related fields exist, and believe cross-contamination across these is auspicious. In terms of nomenclature, we adopted the term "assessor" following [1], because that terminology precedes the one used for anticipatory versions of rejectors considered in [2], and the emphasis of our paper on evaluation.
>
> - **General consideration:**
> We feel that part of the criticism comes from a discussion about overlapping areas (and varying terminology: assessors, rejectors, etc.) that may have obscured the distinctive problem we are addressing, independently of the name: whether we can anticipate the performance of new cases and models having seen only a few examples, and what this means for the evaluation of LLMs. This is novel and we don’t know of previous work to compare with in the same terms.
>
> Considering the explanations above, we kindly ask the reviewer to revise their score if they feel our response adequately addresses their comments, or otherwise to indicate ways in which we can further address them and improve our paper.
>
> [1]: José Hernández-Orallo, Wout Schellaert, and Fernando Martı́nez-Plumed.(2022) ”Training on the test set: Mapping the system-problem space in AI”. In Proceedings of the AAAI conference on artificial intelligence. https://ojs.aaai.org/index.php/AAAI/article/view/21487
>
> [2]: Kilian Hendrickx, Lorenzo Perini, Dries Van der Plas, Wannes Meert, and Jesse Davis. (2024) “Machine
> learning with a reject option: A survey” Machine Learning. https://link.springer.com/article/10.1007/s10994-024-06534-x
>
> [3]: Shnitzer, T., Ou, A., Silva, M., Soule, K., Sun, Y., Solomon, J., ... & Yurochkin, M. (2023). Large language model routing with benchmark datasets. arXiv preprint arXiv:2309.15789.
>
> [4] Drapal, P., Silva-Filho, T., & Prudêncio, R. B. (2024, June). Meta-Learning and Novelty Detection for Machine Learning with Reject Option. In 2024 International Joint Conference on Neural Networks (IJCNN) (pp. 1-8). IEEE.

---

> > ### Comment · Reviewer_XEPY · 2024-11-16
> >
> > I thank the authors for their detailed and thoughtful response. However, it does not address my foundational concerns.
> >
> > Unlike the authors, I see no unique research value in assessor models. The 5 desiderata in https://ojs.aaai.org/index.php/AAAI/article/view/21487 carve out a research domain so narrow that it is hard to see why it is necessary, perhaps explaining why there have been so few follow-up works.
> >
> > It is always possible to design a set of requirements stringent enough that no published work exists to compare against. But that choice comes at a cost. The purpose of this venue is to publish works that have the potential to be high-impact. Without comparison to strong existing work, it is hard to see how this meets that standard, since there are many other ways to accomplish the underlying goal, predicting LLM performance on unseen instances.
> >
> > Furthermore, in this case, even if there are no published baselines, it would have been trivially easy to invent some (e.g., adapting dataset-scale predictors like https://arxiv.org/abs/2402.14992 to instance-scale ones using the average probability over all instances in a scenario, or using an independently trained LLM in a few-shot manner to predict whether the LLM's latest output was correct or not).
> >
> > Therefore, I am keeping my score.

---

> > > ### Author Response · Authors · 2024-11-18
> > >
> > > We thank the reviewer for their comment.
> > >
> > > We believe that the research question determined by the 5 desiderata in that paper is relevant: this amounts to anticipating how a general-purpose AI system such as an LLM will behave for a new input, which is an important research problem. Independently of how this is named, this has multiple applications beyond evaluation, such as implementing reject rules and routing algorithms, to name just two examples, especially in situations where producing the output is costly (e.g., models such as O1, where thinking compute is variable) or in agential models where it is crucial to determine if the next command is to be sent to the agent. This is why our paper is especially relevant now: as part of this overall framework, we explore whether selecting 100 instances randomly and only looking at the input of the new instance yields good predictive power. This has implications and insights for future research, especially in the design of evaluation methods, reject rules and routing mechanisms for chain-of-thought and agential LLMs.
> > >
> > > Regarding baselines, we acknowledge that a few-shot use of an LLM could have been explored and we’ll do so in the camera-ready version if our paper gets accepted. Similarly, we’ll also consider the average probability of success on the benchmark the input prompt belongs to, even though this requires that the user knows what benchmark a new task belongs to, something that is unrealistic in ecologically valid uses of LLMs

---

### Official Review · Reviewer_UgCA · 2024-11-03

**Soundness:** 2
**Presentation:** 2
**Contribution:** 2
**Rating:** 5
**Confidence:** 4

**Summary:**

This paper proposes a method to predict the performance of a new LLM on individual task instances using only a small set of reference instances (approximately 100). The authors introduce a framework that leverages existing evaluation results from multiple LLMs to build a generic assessor. This assessor combines the new LLM's performance on the reference instances with intrinsic features of the task instances to predict performance on new instances. The method is evaluated on two datasets: HELM-Lite and KindsOfReasoning collection, which covers various reasoning tasks. Experiments involve several OpenAI models. The findings suggest that the generic assessor can predict the performance of new LLMs with accuracy comparable to a specific assessor trained on full datasets in in-distribution scenarios. However, the predictive power declines significantly in OOD settings, indicating limitations in generalizing across different data distributions.

**Strengths:**

* **Relevance to Current Challenges:** The paper addresses a practical and pressing issue in NLP—efficiently predicting LLM performance without extensive evaluation on large datasets.

* **Innovative Framework:** Introduces a novel framework that combines a small set of reference evaluations with instance-specific features to build a generic assessor capable of predicting LLM performance on new instances.

**Weaknesses:**

* **Limited Novelty and Contextualization:** The proposed method shares similarities with existing approaches, such as benchmark subsampling strategies and TinyBenchmarks. The paper does not sufficiently compare with or differentiate itself from these methods, which limits the perceived originality.
* **Insufficient Analysis of OOD Performance:** While the significant drop in predictive performance in out-of-distribution scenarios is acknowledged, the paper lacks a thorough analysis of the causes and potential solutions to mitigate this issue.
* **Methodological Clarity:** Some methodological details are not fully elaborated, such as the specific algorithms and parameters used for clustering and factor analysis in reference instance selection. The criteria for choosing hyperparameters and classifiers are also not described in depth.
* **Reliance on Proprietary Features:** The use of OpenAI embeddings as instance-specific features may limit the generalizability and accessibility of the method, particularly for those without access to these embeddings.
* **Evaluation Scope:** The experiments focus on tasks with binary correctness metrics. The applicability of the method to tasks with more complex or continuous evaluation metrics is not explored, which limits the understanding of its generality.
* **Insufficient Quantitative Comparison with Baselines:** The paper lacks detailed quantitative comparisons with existing methods, making it difficult to assess the advantages or disadvantages of the proposed approach in terms of predictive accuracy and computational efficiency.

**Questions:**

1. **Comparison with Existing Methods:** How does your method quantitatively compare with existing approaches like TinyBenchmarks or IRT-based methods in terms of predictive accuracy and computational efficiency? Including such comparisons would strengthen the paper and clarify its contributions relative to prior work.
2. **Improving Out-of-Distribution Performance:** Have you considered techniques to enhance the predictive performance in out-of-distribution scenarios, such as incorporating domain adaptation methods or training on more diverse datasets? A deeper analysis of the factors contributing to the OOD performance drop would be beneficial.
3. **Exploration of Alternative Instance Features:** Did you experiment with other types of instance-specific features that are publicly available and model-agnostic, such as linguistic or syntactic properties? Exploring alternative features could improve generalizability and reduce reliance on proprietary embeddings.

---

> ### Author Response · Authors · 2024-11-16
> **General response to reviewer's comments**
>
> We thank the reviewer for their comments and suggestions. The main points of the reviewer are:
> - Our approach has limited novelty with respect to related approaches: we believe this is not the case, as our approach is original in providing predictions for novel LLM-prompt combinations. Moreover, the reviewer themselves identifies the novelty of our framework as a strength of our work.
> - we should add comparisons with related baselines: we are unaware of any possible baseline method that is directly applicable in our scenario; those mentioned by the reviewer in their question 1 are unsuitable.
> - We should explore the causes of OOD performance drop: we believe this goes out of the scope of our paper, which aims to build a framework leveraging performance results across various LLMs to more efficiently predict performance of new LLMs on individual test prompts. The OOD performance drop is shared by previously introduced methods, thus making it not specific to our approach.
>
>  We provide detailed responses to all identified weaknesses and questions in separate comments below. We believe our responses address the reviewer’s comments, and we thus ask the reviewer to please revise their score if they agree, or otherwise to indicate ways in which we can further address them and improve our paper.

---

> ### Author Response · Authors · 2024-11-16
> **Response to weakness 1**
>
> > Limited Novelty and Contextualization: The proposed method shares similarities with existing approaches, such as benchmark subsampling strategies and TinyBenchmarks. The paper does not sufficiently compare with or differentiate itself from these methods, which limits the perceived originality.
>
> We agree that our method shares some similarities to TinyBenchmarks, particularly when the IRT method by TinyBenchmarks is used to select the reference instances, instead of one of the other two approaches, and when clustering is employed. However, as we discussed in Section 2.4, we stress that the goal of TinyBenchmarks, as well as that of other benchmark subsampling approaches, is fundamentally different from ours: we aim to predict LLM performance on individual prompts, while those works focus on estimating average score on a benchmark by testing on a reduced number of samples.
> In particular, we employ "intrinsic" (model-agnostic) features of the prompt in the performance predictor (alongside the performance on the reference samples, see Fig 1), which makes our trained predictor applicable to novel combinations of LLMs and prompts (thus being applicable to prompts for which the ground truth is unavailable). In contrast, TinyBenchmark uses quantities obtained from the performance of the considered _train_ LLMs to estimate the overall performance. As such, even if their IRT method can in principle estimate performance on individual prompts, it **cannot** do so for novel prompts on which the train LLMs were not evaluated.
>
> We amended Section 2.4, which already contained an extensive comparison between TinyBenchmark and our approach, to make the differences clearer. We kindly ask the reviewer to check the revised section and let us know if this addresses their concerns, and raise their score if this is the case.

---

> ### Author Response · Authors · 2024-11-16
> **Response to weakness 2**
>
> > Insufficient Analysis of OOD Performance: While the significant drop in predictive performance in out-of-distribution scenarios is acknowledged, the paper lacks a thorough analysis of the causes and potential solutions to mitigate this issue.
>
> We agree that our paper does not analyse the causes of the drop in OOD predictability and suggest potential solutions. However, our work focuses on building a framework leveraging performance results across various LLMs to more efficiently predict performance of new LLMs on individual test prompts. The drop in OOD performance is common to previously introduced "specific assessors", and as such we believe a careful study of this to be out of the scope of our work.
>
> For instance, it would be possible to analyse instance-level performance to identify subsets of OOD benchmarks where the model behaves better than others and thus understand what elements make the benchmark OOD. However, we believe that carefully doing this constitutes a separate research work and would excessively expand the scope of our paper.

---

> ### Author Response · Authors · 2024-11-16
> **Response to weakness 3**
>
> > Methodological Clarity: Some methodological details are not fully elaborated, such as the specific algorithms and parameters used for clustering and factor analysis in reference instance selection. The criteria for choosing hyperparameters and classifiers are also not described in depth.
>
> The clustering algorithm we use (KMeans) is extremely standard, and so is factor analysis, to which we still dedicate a full paragraph in Sec 3.2.1. We use the default parameters of commonly used libraries in our implementation, which we will release to ensure complete reproducibility.
>
> The only hyperparameters to be selected in our framework are
> - the number of reference instances, the process to select which is given in Appendix C
> - the intrinsic features, the selection of which is discussed in Sec 4.3
> - the base classifier: as discussed in Sec 4.3, we use different base classifiers with different hyperparameter values and select the best one according to performance on a validation split.
>
> We hope this clarification alleviates the reviewer's concern. Otherwise, can they please point out what specific details they would like to see included in the paper?

---

> ### Author Response · Authors · 2024-11-16
> **Response to weakness 4**
>
> > Reliance on Proprietary Features: The use of OpenAI embeddings as instance-specific features may limit the generalizability and accessibility of the method, particularly for those without access to these embeddings.
>
> While we used the OpenAI embeddings due to their convenience, our method is not constrained to using them, and any prompt-intrinsic features (such as other embeddings) work. In particular, we stress how the specific assessor reaches similar performance when using the Word2Vec and FastText embeddings. Finally, the OpenAI embeddings are publicly available to anyone who registers an account with OpenAI and computing them has a very small monetary cost (<1$ for the complete datasets we considered). We will also release the computed embeddings publicly.

---

> ### Author Response · Authors · 2024-11-16
> **Response to weakness 5**
>
> > Evaluation Scope: The experiments focus on tasks with binary correctness metrics. The applicability of the method to tasks with more complex or continuous evaluation metrics is not explored, which limits the understanding of its generality.
>
> The reviewer is correct in pointing out that our study focuses on binary correctness metrics. Indeed, our framework (the selection of reference instances and the use of intrinsic features alongside those to predict performance of a new LLM on a novel prompt) can easily be adapted to problems where the score metric is non-binary, by replacing the binary classifiers we used with the suitable regression or classification algorithm. It is thus a valid question to ask whether our findings would generalise in those scenarios, which is a question we leave for future work (we now explicitly added this in the conclusion). In this first work, we chose to focus on binary correctness metrics as most benchmarks use such metrics, thus making our initial investigation more relevant and allowing us to carefully analyse this scenario.

---

> ### Author Response · Authors · 2024-11-16
> **Response to weakness 6**
>
> > Insufficient Quantitative Comparison with Baselines: The paper lacks detailed quantitative comparisons with existing methods, making it difficult to assess the advantages or disadvantages of the proposed approach in terms of predictive accuracy and computational efficiency.
>
> Can the reviewer please suggest examples of existing methods which we could apply in our considered scenario and compare to?

---

> ### Author Response · Authors · 2024-11-16
> **Response to questions**
>
> **Questions:**
> > Comparison with Existing Methods: How does your method quantitatively compare with existing approaches like TinyBenchmarks or IRT-based methods in terms of predictive accuracy and computational efficiency? Including such comparisons would strengthen the paper and clarify its contributions relative to prior work.
>
> As stressed in our response above to the first comment of the reviewer, TinyBenchmarks and other IRT-based methods are inapplicable to novel prompts on which previous models have not been evaluated, while our approach is. Indeed, we aimed to predict performance of a novel LLM on novel questions, so that our approach could be deployed in operational scenarios, where the ground truth of the question may not be known, preventing older LLMs from being tested and evaluated. Therefore, a direct comparison with these two methods is not possible.
>
>
> > Improving Out-of-Distribution Performance: Have you considered techniques to enhance the predictive performance in out-of-distribution scenarios, such as incorporating domain adaptation methods or training on more diverse datasets? A deeper analysis of the factors contributing to the OOD performance drop would be beneficial.
>
> As we mentioned above, addressing the drop in OOD predictability is out of the scope of this work, which focuses on building a framework leveraging performance results across various LLMs to more efficiently predict performance of new LLMs on individual test prompts.
>
> Moreover, training on more diverse datasets would not address the root cause for the drop in OOD predictability but rather render the test dataset "less OOD". In terms of domain adaptation methods, if the reviewer means re-calibrating the trained assessor on the OOD dataset, we point out that this requires labelled data from the latter, which may not be representative of real-world use cases.
>
> > Exploration of Alternative Instance Features: Did you experiment with other types of instance-specific features that are publicly available and model-agnostic, such as linguistic or syntactic properties? Exploring alternative features could improve generalizability and reduce reliance on proprietary embeddings.
>
> We stress that all instance features that we employ are agnostic to the considered set of LLMs. Moreover, we employed n-grams, Word2Vec and FastText in the experiments for the specific assessor.

---

> > ### Comment · Reviewer_UgCA · 2024-11-28
> >
> > I thank the authors for their detailed and thoughtful response. While they have clarified some of my concerns, such as the differences against related works and the hyperparameter choices, several fundamental issues remain unaddressed.
> > As other reviewers have noted, my main concern is the limited scope of the method. The authors have designed an approach with such strong requirements that it raises questions about appropriate baselines for comparison and the broader impact of the contribution. Additionally, the generalization of the method to non-binary score metrics remains an open question.
> > Based on these ongoing concerns, I maintain my original score.

---

### Official Review · Reviewer_1jCJ · 2024-11-04

**Soundness:** 3
**Presentation:** 3
**Contribution:** 2
**Rating:** 5
**Confidence:** 3

**Summary:**

This paper explores a framework for predicting the performance of large language models (LLMs) on new tasks by using a generic assessor based solely on observational features of the dataset instances. With the rapid release of new LLM versions, traditional methods for evaluating their correctness can be resource-intensive. The authors propose characterizing each LLM by its performance on a small set of reference instances, allowing for efficient predictions without extensive evaluations. Their studies suggest that this generic assessor performs comparably to specific assessors within the same task distribution, though its effectiveness declines in out-of-distribution scenarios. Additionally, the research introduces the "KindsOfReasoning" dataset to support evaluations of reasoning capabilities across various models. Overall, the study provides insights into LLM validation and the potential for more streamlined assessment methodologies.

**Strengths:**

- The paper presents a novel framework for predicting LLM performance using a generic assessor, addressing a critical challenge in the fast-evolving landscape of large language models.
-Empiric validation of the proposed methodology demonstrates that the generic assessor can achieve results comparable to specific assessors, offering a more efficient alternative for performance evaluation.
- The research contributes to the understanding of generalizability by highlighting the limitations of predictive power in out-of-distribution scenarios, prompting further investigation into LLM behavior across diverse tasks.
- The introduction of the "KindsOfReasoning" dataset enhances the available resources for future research, providing a valuable compilation of reasoning tasks that can stimulate further exploration in LLM capabilities.

**Weaknesses:**

- L340: “To the best of our knowledge, this is the first collection of instance-level results covering all versions of a given model family”. I believe this claim should be softened. There’s quite the history of releasing completions for models before (LiveBench and numerous huggingface repositories for example). This doesn’t deminsh the importance of your releasing these completions — I think that’s great — but it isn’t as strong of a separator of your work as you may be thinking.
- I believe a fuller description of “KindsOfReasoning” should be included in the main body of the paper. I don’t think it is sufficient to just describe it as _comprised of reasoning tasks for existing datasets_. I would suggest at least a paragraph or two about its construction. Also Appendix B does not include any reasoning for how these data were included and other data excluded from the set, and important consideration. Finally, I’d like to learn more about what validation you did about the questions, prompts, etc to ensure the collection of data were covering topics of interest, not duplicative, and stylistically aligned. You could pick up space here by removing some of the more well-known and basic parts of machine learning which you explain in great detail: train-val-test splits, AUC, etc.
- “the latter two generate a vector for each word in the prompt, which we average to form a vector representing the entire prompt” this seems like it removes a lot of information. Can you talk about other methods you could have or did try and why this approach was taken?
- L422: “pick the one with the highest value”. I have a serious concern with this. Can you describe why this is the approach taken and not reporting each individual model performance, as I think is much more standard int he community? I realize it may have to do with how you envision this approach working in practice, but please explain this (to me) non-standard model-selection method given my understanding of the task at hand.


Minor points:
- (This is slightly more significant than “minor”.) I think the use of the term “instance” is somewhat non-standard in the entirety of the community. As I understand it, you mean “instance” to refer to a (prompt, answer) pair. If this is the case, (and even if it isn’t), I suggest defining this term very early in the introduction (maybe even abstract) to make it clear to the diverse set of readers of your paper.
- I suggest splitting the paragraph starting at L159 into several smaller paragraphs for easier reading. Perhaps at L166: “Closer to our work…” and L178: A similar work to Polo…”.

Typos:
- L120: “use cases.Following”
- L428: I think you want \citep not \citet (parenthetical cite for Kuspati et al)
- L468: simpler -> simple

**Questions:**

See weaknesses

---

> ### Author Response · Authors · 2024-11-16
> **Response to reviewer's comments**
>
> We thank the reviewer for their comments and suggestions. We discuss the major points below. We also updated our manuscript following some of the reviewer's suggestions. In particular, we notice that the reviewer did not point out any weakness related to our framework or its empirical validation. Indeed, they highlighted as strengths the novelty and efficiency of our framework, the insights provided by our experiments, and the usefulness of the KindsOfReasoning dataset for future research. In light of this, the clarifications provided below and the amendments made to the text, we invite the reviewer to raise their score.
>
> > L340: “To the best of our knowledge, this is the first collection of instance-level results covering all versions of a given model family”. I believe this claim should be softened. There’s quite the history of releasing completions for models before (LiveBench and numerous huggingface repositories for example). This doesn’t deminsh the importance of your releasing these completions — I think that’s great — but it isn’t as strong of a separator of your work as you may be thinking.
>
> We thank the reviewer for pointing that out; we have now removed that claim.
>
> > I believe a fuller description of “KindsOfReasoning” should be included in the main body of the paper. I don’t think it is sufficient to just describe it as comprised of reasoning tasks for existing datasets. I would suggest at least a paragraph or two about its construction. Also Appendix B does not include any reasoning for how these data were included and other data excluded from the set, and important consideration. Finally, I’d like to learn more about what validation you did about the questions, prompts, etc to ensure the collection of data were covering topics of interest, not duplicative, and stylistically aligned.
>
> Thanks for this suggestion, we added the following  paragraph in Section 4.1:
>
> The datasets were selected to cover a wide range of kinds of reasoning (logical, common sense, inductive, deductive, abductive, counterfactual, causal, analogical, spatial and arithmetic reasoning). In particular, we conducted a keyword search in known benchmark repositories (BIG-Bench and HELM) and academic search engines for benchmarks about reasoning. Of those we found, we excluded those that require a large amount of commonsense knowledge (such as SocialIQA), test the dependence of reasoning abilities on context (such as NeuBAROCO) or whose license did not allow derivative works to be distributed (ART). The final collection contains datasets with different prompting styles, as true reasoning abilities should be robust to these variations.
>
> > “the latter two generate a vector for each word in the prompt, which we average to form a vector representing the entire prompt” this seems like it removes a lot of information. Can you talk about other methods you could have or did try and why this approach was taken?
>
> The reviewer is here referring to our use of the Word2Vec and FastText embeddings, which are word-level embeddings, and they are correct in saying that the aggregation discard information. Indeed, our motivation for including those embeddings was to understand whether sentence embeddings, such as the OpenAI embeddings we employed, provided benefits compared to average word-level embeddings.
>
> > L422: “pick the one with the highest value”. I have a serious concern with this. Can you describe why this is the approach taken and not reporting each individual model performance, as I think is much more standard int he community? I realize it may have to do with how you envision this approach working in practice, but please explain this (to me) non-standard model-selection method given my understanding of the task at hand.
>
> We apologise for the confusion; what we are employing there is a traditional train-validation-test split procedure: for a fixed set of features (eg, Word2Vec) and LLM, we train several base classifiers on the training data. We then use the performance on the validation data to select the best base classifier (analogous to selecting hyperparameters) and then report the performance of the classifier with the best validation performance on the test data, to avoid biasing the test performance due to the selection. We amended the text to clarify this.
>
>
> > (This is slightly more significant than “minor”.) I think the use of the term “instance” is somewhat non-standard in the entirety of the community. As I understand it, you mean “instance” to refer to a (prompt, answer) pair. If this is the case, (and even if it isn’t), I suggest defining this term very early in the introduction (maybe even abstract) to make it clear to the diverse set of readers
>
> We thank the reviewer for pointing out this potential cause for confusion. The term “instance” is commonly used in this sense in psychometrics, but we agree that it is not common in AI or ML. We clarified the meaning of the term in the abstract and the introduction.

---

> > ### Comment · Reviewer_1jCJ · 2024-11-26
> >
> > Thank you for your work and response to my concerns! I appreciate the updated PDF and see the effort to improve the paper. I believe that the paper is improved, and I have raised my score accordingly.

---

### Author Response · Authors · 2024-11-16
**General answer to reviewers' comments**

The first two reviewers stress the novelty of our framework for predicting the LLM performance by only evaluating on a few instances, which they state is addressing a relevant problem in an efficient fashion. Moreover, the first and third reviewer highlight the usefulness of our newly introduced metadatasets (KindsOfReasoning) to future research, and the insights our empirical studies shed into generalisability of LLM predictability in OOD scenarios.

The reviewers also stress the following common weaknesses:
- Reviewers 2 and 3 claim that we should conduct a better comparison with related fields of research, both in the discussion and in terms of baseline methods to include in our experimental studies. In our responses, we clarified that the originality of our work comes from predicting the performance of a novel LLM on a novel prompt. As such, even if several related methods exist, we are unaware of approaches that could be applied in our setup. We asked the reviewers to advise on concrete baseline methods which we could compare to.
- Reviewers 1 and 3 claim that the contribution of KindsOfReasoning is limited by being a metadataset: we agree that this is the case, but 1) introducing that was not the main focus of our work, and 2) we provide instance-level results for several LLMs on those, which can help further research, as highlighted by the reviewers themselves.

We respond to the other weaknesses indicated by individual reviewers in separate comments.

We also amended the manuscript following reviewers’ comments. A tracked-changes version is available in the supplementary material.

---

### Meta-Review · Area_Chair_NYwn · 2024-12-19

**Metareview:**

The submission proposes a workflow for evaluating LLMs based on training an assessor model to extrapolate validation scores from a small set of instances the the LLM has been evaluated on directly. A highlight of the findings is that it is found that only 100 instances are needed to estimate the performance of LLMs within this framework. However, they note that the assessor models they train suffer for poor robustness to out-of-distribution scenarios.

The reviewers suggest that the novelty and significance of the proposed approach and associated meta-dataset are insufficient to warrant acceptance of the submission. Moreover, reviewer XEPY suggests that baseline methods should be compared with.

**Additional Comments On Reviewer Discussion:**

The main outcome of the discussion is that reviewers conclude that the authors have restricted the scope of their study so much that it is not possible to compare with other methods.

---

### Decision · Program_Chairs · 2025-01-22

Reject